# The 2-D Cluster Variation Method: Topography Illustrations and Their Enthalpy Parameter Correlations

**DOI:** 10.3390/e23030319

**Published:** 2021-03-08

**Authors:** Alianna J. Maren

**Affiliations:** 1Data Science Faculty, Northwestern University School of Professional Studies, Evanston, IL 60208, USA; alianna.maren@northwestern.edu; 2Themasis Associates, P.O. Box 274, Kealakekua, HI 96750, USA; alianna@aliannajmaren.com

**Keywords:** cluster variation method, entropy, approximation methods, free energy, free energy minimization, statistical thermodynamics, topography

## Abstract

One of the biggest challenges in characterizing 2-D image topographies is finding a low-dimensional parameter set that can succinctly describe, not so much image patterns themselves, but the nature of these patterns. The 2-D cluster variation method (CVM), introduced by Kikuchi in 1951, can characterize very local image pattern distributions using configuration variables, identifying nearest-neighbor, next-nearest-neighbor, and triplet configurations. Using the 2-D CVM, we can characterize 2-D topographies using just two parameters; the activation enthalpy ($\varepsilon_{0}$) and the interaction enthalpy ($\varepsilon_{1}$). Two different initial topographies (“scale-free-like” and “extreme rich club-like”) were each computationally brought to a CVM free energy minimum, for the case where the activation enthalpy was zero and different values were used for the interaction enthalpy. The results are: (1) the computational configuration variable results differ significantly from the analytically-predicted values well before $\varepsilon_{1}$ approaches the known divergence as $\varepsilon_{1}\to 0.881$, (2) the range of potentially useful parameter values, favoring clustering of like-with-like units, is limited to the region where $\varepsilon_{0}<3$ and $\varepsilon_{1}<0.25$, and (3) the topographies in the systems that are brought to a free energy minimum show interesting visual features, such as extended “spider legs” connecting previously unconnected “islands,” and as well as evolution of “peninsulas” in what were previously solid masses.

## 1. Introduction

As a complement to structural topology descriptions, the 2-D cluster variation method (CVM) can be viewed as a generative means for producing different topologies. The topographies generated by the 2-D CVM can be characterized by just two parameters; an activation enthalpy ($\varepsilon_{0}$) and an interaction enthalpy ($\varepsilon_{1}$). As an initial 2-D grid pattern is brought towards an equlibrium state for a specific pair of enthalpy values ($\varepsilon_{0},\varepsilon_{1}$) (using the 2-D CVM free energy formulation), the resulting topography can be characterized by a set of configuration variables. This topography also evinces certain characteristics associated with the selected enthalpy parameters.

In this study, for which the 2-D CVM grid is constructed as a a juxtaposition of multiple, overlapping zigzag chains, the configuration variables are nearest and next-nearest neighbors, as well as triplets. Thus, any specific initial topography, undergoing free energy minimization for a specific pair of entropy parameters, moves towards a resulting equilibrium-state topography. Both the initial and resulting topographies can be characterized by their associated configuration variabiles, such as the fraction of triplets in a given state; e.g., **A**-**A**-**A**, where the **A** units are “on” and the **B** units are “off.” (Throughout this work, the terms “unit” and “node” are used interchangeably.)

This work addresses three questions, the answers to which will move forward the feasibility of using the 2-D CVM for characterizing topographies that can be described using a 2-D grid. This will then suggest pathways for *topography evolution*, as the governing enthalpy parameters are changed. From this, applications to natural and abstract topographies can be investigated.

The three questions investigated here are:How do the configuration variables associated with at-equilibrium patterns compare with the analytically-predicted configuration variables, for the particular set of enthalpy parameters $\left(\varepsilon_{0},\varepsilon_{1}\right)$ used to drive the initial pattern towards a free energy minimum?What is the range of a useful $\left(\varepsilon_{0},\varepsilon_{1}\right)$ phase space, and (as an extension of question 1), what are the configuration variable values associated with points in this space? andHow can we visually characterize the resulting topographies themselves; that is, can we identify a correspondence between observed visual patterns and associated configuration variables in an at-equilibrium 2-D grid system?

To ground this discussion, the following Section 1.1 presents a worked example, showing both an initial grid pattern and the resultant pattern, after the system has been brought to equilibrium for a specific set of enthalpy parameters ($\varepsilon_{0},\varepsilon_{1}$).

To facilitate correspondence between computationally-achieved results with the analytic predictions, all trials presented here are for the case where the activation enthalpy parameter $\varepsilon_{0}=0$; this is the case where an analytic solution is possible.

### 1.1. A Worked Example: Instance of a Resulting 2-D CVM

The following Figure 1 illustrates the sort of characterization afforded by the 2-D CVM approach. The remainder of this paper will unpack the construction and interpretation of this figure, to provide a worked example of how the 2-D CVM approach is applied.

Figure 1 illustrates an example of applying the 2-D CVM to a 256-element (16 × 16) grid. A manually-designed initial system on the LHS (Left-Hand-Side) has been brought to free energy equilibrium on the RHS (Right-Hand-Side).

The remaining sections will investigate this figure in more depth; this is presented here as an illustration of the kind of results obtained via free energy minimization in a 2-D CVM grid.

After the initial, manually-constructed topography (shown in Figure 1a) has been brought to a free energy minimum, the resulting topography (shown in Figure 1b), has a very different appearance, and has associated with it a substantially different set of configuration variables. This particular topographic result was accomplished with only a single non-zero enthalpy parameter. The figure was designed with equal numbers of “on” (state **A**, shown in black) and “off” (state **B**, shown in white) units. This corresponds to the case where the unit activation enthalpy was set to zero; $\varepsilon_{0}=0$. The only remaining parameter was the interaction enthalpy between a pair of neighboring units, which was here set to a fairly high value of $\varepsilon_{1}=0.250$, or (more useful in the actual computations) h=1.65, where the *h-value* (or simply *h*) is given as $h=exp(2\varepsilon_{1})$.

The manually-designed pattern used as a starting point in Figure 1a was a “scale-free-like” design, in that it had masses of “on” (**A**) units enbedded into a sea of “off” (**B**) nodes. The original design plan was to have masses of of 16, 8, and 4 units, with increasing numbers of units as the mass-size decreased. This original plan had to be adjusted by the constraint keeping equal numbers of (**A**) and (**B**) units. This constraint kept the unit activation enthalpy at zero ($\varepsilon_{0}=0$), which allowed direct comparison of the computational configuration variable results with those obtained analytically. The configuration variables are fractional values for nearest-neighbor, next-nearest-neighor, and triplet configurations. The cluster variation method rests on including these variables in the entropy term of the free energy formulation.

One of the interesting things about the 2-D CVM approach is that we can see immediate visual results corresponding to our selection of the interaction enthalpy parameter, $\varepsilon_{1}$. The value used here, h=1.65 ($\varepsilon_{1}=0.25$), strongly encouraged pairwise interactions. Had free energy minimization continued (using a more sophisticated node state adjustment algorithm), the resulting pattern could have moved much more strongly in the direction of a “rich club” configuration. Even with existing algorithmic constraints, we see that previously unconnected “islands” of (**A**) (black nodes) in a sea of (**B**) (white) are now connected, typically via extended spider leg-like diagonal trails of black (**A**) nodes, connecting the previously-isolated **A**-masses. This is just one of several observations that we can make about topographies resulting from 2-D CVM free energy minmization.

The following subsections introduce the 2-D CVM, with specific attention to the configuration variables, which provide very local topographic descriptive elements. These ensuing subsections also briefly review the evolution of the CVM and selected applications.

### 1.2. Introducing the Cluster Variation Method

The cluster variation method was originally devised by Kikuchi in 1951 [1], and then further advanced by Kikuchi and Brush (1967) [2]. Maren (2016) suggested that the CVM would be appropriate for modeling brain topographies [3], and presented initial computational results for the 1-D CVM. More recently, Maren (2019a) investigated how bringing a system towards an equilibrium state influenced resultant 2-D topographies represented by a 2-D CVM grid [4].

The essential notion of the CVM is that we work with a more complex entropy expression within the free energy formalism for a system.

In a simple Ising model, the entropy *S* can be computed based on only the relative fraction of active units in a bistate system. That is, there are only two kinds of units; *active* ones in state **A**, where the fraction of these units is denoted $x_{1}$, and *inactive* ones in state **B**, where the fraction of these units is denoted $x_{2}$. (Of course, $x_{1}+x_{2}=1.0$.)

In contrast to the simple entropy used in the basic Ising model, in the CVM approach, we expand the entropy term. The CVM entropy term considers not only the relative fractions of units in states **A** and **B**, but also a set of configuration variables. The configuration variables for a 2-D CVM system include, in addition to the usual activation of single units, also the fractional values of different kinds of nearest neighbor and next-nearest-neighbor pairs, along with six different kinds of triplets. (Section 2.1 describes the configuration variables in depth.)

Our entropy term now involves what are essentially *topographic* variables. That is, the location of one unit in conjunction with another now makes a substantial difference.

As a result, when we do free energy minimization to find an equilibrium configuration for a given system, we arrive at a 2-D CVM grid with certain characteristic topographic properties. While the exact activation of a specific unit (whether it is in state **A** or state **B**) can vary, each time a given system is brought to equilibrium, the overall values for the configuration variables should (ideally) be consistent for a given set of enthalpy parameters.

In practice, there is likely to be a range of resulting configuration variable values, depending on initial topographies and the number of steps taken towards free energy minimization. The topographic sophistication of the free energy minimization algorithm will also play a role; the one used for this work is very simple. Thus, the focus of this work is on characterizing the topographic patterns, and their concommitant configuration variable values, associated with movement towards equilibrium states at specific enthalpy values.

### 1.3. Background: The Cluster Variation Method (CVM)

The cluster variation method (CVM) was originally devised by Kikuchi in 1951 [1], and then further advanced by Kikuchi and Brush (1967) [2]. The CVM is actually a hierarchy of approximate variational methods, applied to Ising models in equilibrium statistical physics. Over the past two decades, various researchers have placed the CVM into a broader context, most notably within graph theory and belief propagation methods.

One of the most interesting evolutions in CVM conceptualization has been that it can be used for belief propagation, as proposed by Yedidia, Weiss, and Freeman in 2001 [5,6]. They have shown that, if they defined the messages and message-update rules appropriately, the fixed points of a generalized belief propagation (GBP) algorithm were equivalent to the stationary points (equilibrium points) of the corresponding CVM approximation [7]. (Yedidia (2000) has also provided a pleasant introduction to the CVM in the context of an overall discourse on Gibbs free energy, and (more specifically) the Ising equation [8].)

Pelizzola (2005) extensively reviewed the CVM [9], with a substantial discussion of how the CVM relates to probabilistic graph models and message passing algorithms. Wainwright and Jordan (2008) then described the CVM within the larger scope of graphical models, exponential families, and variational methods [10].

The majority of CVM studies have been with 3-D systems, most notably in applications to alloys [11]. Albers et al. (2006) applied the CVM to efficient linkage analysis [12], and Barton and Cocco (2013) studied Ising models for neural activity inferred via selective cluster expansion [13]. Recently, Balcerzak and Szalowski (2020) have formulated the pair approximation method (a CVM extension) for the isotropic ferromagnetic Heisenberg model with spin S = 1, accounting for both single-ion anisotropy and an external magnetic field [14].

There have been only a few studies of the 2-D CVM. Dominguez et al. (2015) have compared a Bethe and plaquette-CVM for the random field Ising model in two dimensions, and showed that in both methods, it was difficult to find a a robust critical line [15].

While there have been these substantive efforts to place the CVM within the broader context of graphical systems and methods (such as belief propagation) that can be used in such systems, to this author’s knowledge, there has not yet been investigation into the *actual topographies* produced by the CVM during the course of free energy minimization, applied to an initial landscape or grid pattern.

As a precursor to the current work, Maren (2016) used a 1-D CVM architecture where the fundamental units were nearest-neighbors, next-nearest neighbors, and triples [3]. This present work carries that earlier investigation forward, focusing on the 2-D case.

The 2-D CVM grids shown in this work follow one of the options first discussed by Kikuchi and Brush (1967) [2]. This option is to use a grid composed of zigzag chain overlays, so that units in one row are offset by those in the above and below rows by a half-unit. This configuration yields a set of configuration variable values where the nearest neighbors are in the rows above and below a given unit, and the next-nearest neighbors are in the same row (visually, immediately adjacent to the central unit), and in two rows above and two rows below. Triplets are also included in this composition.

The reasons for selecting this composition are twofold, and are based on the potential for using the 2-D CVM in application to other endeavors:A grid pattern where a key variable is a configurational triplet has at least some possiblity of being correlated with existing cluster description methods, e.g., the *clustering coefficient* measure first devised by Watts and Strogatz (1998) [16], andA triplet configuration is an appropriate modeling unit for neural systems; see discussion in Section 2.3 of Maren (2016) [3] and more specifically in Sporns and Kötter (2004), who note that neural motifs (or functionally- and structurally-connected units) on the order of M = 3 (where M is the number of units) are a useful modeling base [17].

The following Table 1 presents a glossary of the thermodynamic terms used in this article.

In what follows, this work will use numerical analyses of synthetic activation patterns on a two-dimensional grid as they evolve to–and potentially attain–an equilibrium steady-state distribution. The focus of these analyses is on the characterisation of various patterns in terms of configuration variables; namely, the statistics of local patterns of co-activations or dynamical motifs. By equipping configuration variables with an activation and interaction enthalpy, one can cast evolution to equilibrium as a result free energy minimization.

To simulate this process, one can randomly exchange activation values between grid units and retain the switch if free energy falls. The focus of this paper is on the different kinds of patterns (specifically here, an “extreme rich club-like” and a “scale-free-like” pattern) that emerge under different contributions of activation and interaction enthalpy to free energy. In summary, we will use worked (numerical) examples to illustrate how different kinds of patterns emerge under the same free energy minimizing formalism–and discuss how this may have implications for self-organization and message passing on graphs, such as the brain.

## 2. Results

There are three key results:Deviation of computational results from the analytic,Identification of a useful parameter range, andIntial topography characterization.

In order to present these results, it is useful to have further description of the configuration variables and how they interact with each other. Thus, this Results section is divided into five parts:The configuration variables,Overview of experimental trials,Result 1: computational configuration variable values differ from analytic,Result 2: identification of a useful parameter range, andResult 3: initial topography characterization.

### 2.1. The Configuration Variables

The key requirement to working with the cluster variation method is that we need to understand and use a set of configuration variables. In a simple Ising system (the simplest statistical mechanics model for a system, in which only two states are possible for each unit), we only need to think about whether a given unit is “on” or “off,” or in states **A** or **B**.

We really only need one variable to describe the distribution of units in such a system, because if we let $x_{1}$ be the fraction of nodes in state **A** (“on,” and shown as black units in the 2-D CVM grid), and $x_{2}$ be the fraction of nodes in state **B** (“off,” the white units), then we also have that $x_{2}=1-x_{1}$. This gives us a system that is relatively easy to model and visualize.

In contrast to a simple Ising system, in which we only need the relative fractions of $x_{1}$ and $x_{2}$ units, in the CVM, we expand the set of variables under consideration. We use a set of variables that are collectively referred to as the configuration variables.

In this section, we address three key factors that are essential for working with the configuration variables:Configuration variable definitions–how they show up in the 2-D CVM grid,Counting the configuration variables–how each configuration variable is counted, and the verification and validation (V&V) thereof, andA brief interpretation–how to interpret configuration variable values.

Later in this paper, we will examine how configuration variables change with the enthalpy parameters. However, even before we connect the grid topographies (expressed via configuration variables) with the free energy equation, we can establish a foundation for understanding what the configuration variables mean (in practical terms), and how they interact with each other.

#### 2.1.1. Introducing the Configuration Variables

A 2-D CVM is characterized by a set of configuration variables, which collectively represent single unit, pairwise combination, and triplet values. The configuration variables are denoted as:$x_{i}$-Single units,$y_{i}$-Nearest-neighbor pairs,$w_{i}$-Next-nearest-neighbor pairs, and$z_{i}$-Triplets.

These configuration variables are illustrated for a single zigzag chain in Figure 2.

For a bistate system (one in which the units can be in either state **A** or state **B**), there are six different ways in which the triplet configuration variables ($z_{i}$) can be constructed, as shown in Figure 3.

Notice that within Figure 3, the triplets $z_{2}$ and $z_{5}$ have two possible configurations each: **A**-**A**-**B** and **B**-**A**-**A** for $z_{2}$, and **B**-**B**-**A** and **A**-**B**-**B** for $z_{5}$. This means that ***there is a degeneracy factor of 2*** for each of the $z_{2}$ and $z_{5}$ triplets. This is shown in Figure 3.

The degeneracy factors $\beta_{i}$ and $\gamma_{i}$ (number of ways of constructing a given configuration variable) are shown in Figure 4. For the pairwise combinations $y_{2}$ and $w_{2}$, $\beta_{2}=2$, as $y_{2}$ and $w_{2}$ can each be constructed as either **A**-**B** or as **B**-**A** for $y_{2}$, or as **B**- -**A** or as **A**- -**B** for $w_{2}$. Similarly, $\gamma_{2}=\gamma_{5}=2$ (for the triplets), as there are two ways each for constructing the triplets $z_{2}$ and $z_{5}$. All other degeneracy factors are set to 1. See the illustrations in Figure 3 and Figure 4.

#### 2.1.2. CVM Topographies: Interpreting Configuration Variables

When we work with a typical Ising system model, we can easily interpret the results in terms of simple dependence of various functions on a single variable, $x_{1}$. The topographic organization of the system, in this simple case, is not important.

In contrast, when we work with a 2-D CVM, the topographic organization is all-important. There are fourteen different configuration variables that collectively describe this topography. To assist our interpretation of the configuration variable values corresponding with a given 2-D grid, it helps to identify a few key variables that, taken together, give us insight into their associated topographies. We will call this reduced subset the *interpretation variables*.

It will be useful if we can mentally correlate this very small set of *interpretation variable values* with some notion of their corresponding topographies. We understand, of course, that a given set of values for the *interpretation variables* will not define a specific topography; rather, they will correlate with a specific *kind of* topology.

We nominate three configuration variables to form our set of interpretation variables: $z_{1}$, $z_{3}$, and $y_{2}$. This reduces the number of configuration variables that we consider from fourteen down to three; that is, we go from the set of two $x_{i}$, three each for the $y_{i}$ and $w_{i}$, and six of the $z_{i}$, down to just two; $z_{i}$ and a single $y_{i}$. As we will see in the remainder of this work, this subset is sufficient to give good insight into the corresponding topographies.

These three *interpretation variables* are selected because, taken together, they are reasonably descriptive of the grid topography:$z_{1}$-**A**-**A**-**A** triplets; indicates the relative fraction of **A** units that are included within the *interiors* of the “islands” or “land masses”; $z_{1}$ also (indirectly) indicates the compactness of these masses,$z_{3}$-**A**-**B**-**A** triplets; indicates the relative fraction of **A** units that are involved in a “jagged” border (one that involves irregular protrusions of **A** into a **B** space), or the presence of one or more thin “rivers” of **B** units extending into landmass(es) of **A** units, and$y_{2}$-**A**-**B** nearest-neighbor pairs; indicates the relative extent to which the **A** units are distributed among the surrounding **B** units. A higher $y_{2}$ value indicates lots of boundary areas between **A** and **B**, and a smaller value indicates more compact “landmasses“ of **A** units.

Clearly, the corresponding values for $z_{6}$ (**B**-**B**-**B**) and $z_{4}$ (**B**-**A**-**B**) will indicate similar topographic aspects for the **B** units.

In order to obtain the next simplifications, and also to obtain an *analytic solution* to the 2-D CVM free energy equation, we introduce the constraint that $x_{1}=x_{2}=0.5$. It is only at this point of equiprobable distrubution of **A** and **B** units that the analytic solution can be found.

The analytic solution was first presented, without the detailed derivation, in Kikuchi and Brush (1967) [2]. Derivation details were given in Maren (2019a) [4].

The following Section 2.2 overviews the set of three different types of experimental trials undertaken in this work. Following that, in Section 2.3: Result 1, we compare the values found for the configuration variables at free energy equilibrium with those obtained analytically, and find that the values diverge from the analytically-predicted values as the interaction enthalpy $\varepsilon_{1}$ increases from zero.

The final two subsections address the useful range for the parameters $\left(\varepsilon_{0},\varepsilon_{1}\right)$ (in Section 2.4) and describe characteristic topographic features that emerge in the at-equilibrium patterns, after free energy minimization has been achieved (in Section 2.5).

### 2.2. Overview of Experimental Trials

To investigate the potential use of 2-D CVM grid patterns for modeling both natural and human-created patterns, the experiments used two different kinds of patterns:Scale-free-like, andExtreme rich club-like.

The experimental objectives were to:Identify appropriate interaction enthalpy parameters that reasonably corresponded to the two different initial grid patterns,Identify the actual configuration variable values for each pattern as each was brought to its respective free energy equilibrium, andCharacterize the resulting free energy-minimized topographies.

Each of the experimental trials began with one of the two manually-designed grid configurations shown in Figure 5. These two configurations corresponded (approximately) to the notions of (a) “rich club” topographies and (b) “scale-free.” (For a discussion of these two topographic types, see Section 3: Discussion.)

These two initial grid configurations were designed, not so much to emulate patterns observed in any specific field (e.g., natural landscapes or structural forms of neural organization; see, e.g., [18]), but rather to provide two starting instances that were polar opposites from each other and which thus would give interesting results after each system is brought to (or towards) an equilibrium state, especially when different initial topographies were brought towards a free energy minimum using the same enthalpy parameters.

The “scale-free” design was created following the notion of “far more small things than large ones,” as described by Jiang and Yin (2014) in their introduction of the *ht-index*, or *head-tail index* for describing a fractal system [19]. Rather than be specifically correct in the relative sizes of various “islands” in the overall grid pattern, the numbers of these islands and their respective sizes were constrained by the need to fit the total number of state **A** units, equal to the total number of state **B** units, into the overall grid. Further discussions of how the patterns were created are given in Section 2.5 and in Section A.6.

As we will see in the following Section 2.3, these two initial grid patterns were each designed to have equal numbers of nodes in states **A** and **B**, that is, $x_{1}=x_{2}=0.5$. This was because, when this equiprobability condition holds for the two states **A** and **B**, there is an analytic solution for the configuration variables as a function of the interaction enthalpy. To make the dependence easier, we will use the term *h-value*, where the *h-value* is a simple exponential function of the interaction enthalpy parameter $\varepsilon_{1}$. The analytic solution is achieved using the *h-value*, and so it is easier to couch all descriptions of analytic and computational results, as well as topographic behaviors, in terms of the *h-values*.

As an indicator of experimental intention, Figure 6 shows the analytic solution for three of the configuration variables ($z_{1}$, $z_{3}$, and $y_{2}$) as functions of the *h-value*. It further identifies how each of the two initial grids correspond to candidate *h-values*; (a) the “extreme rich club-like” initial pattern has configuration variable values indicating that an *h-value* of approximately 1.65 will be useful. In contrast, (b) the “scale-free-like” initial pattern has configuration variable values suggesting that a relatively low *h-value* of 1.16 may be appropriate for beginning free energy minimization.

The results of bringing each of these two different initial topographies towards an equilibrium state are presented in Section 2.4. This leads to identifying useful parameter ranges. Section 2.5 discusses the resulting topographies. Details of how candidate *h-values* were selected are presented in Section 4: Materials and Methods. The following Section 2.3 provides an overview of how the analytic solution is achieved.

From Figure 6, it is clear that we can readily characterize a 2-D grid pattern in terms of a potential range of *h-values*, even if it is not yet at free energy equilibrium. There is a distinction between the two initial topographies and their realm of associated *h-values*.

In the instance of the “extreme rich club-like” topography illustrated in Figure 5a, we see an initial set of configuration variable values (diamonds denoted with “**a**” in Figure 6) that are closely aligned with h=1.65. (Note that this mass of **A** nodes represents a single “continent,” due to wrap-arounds of the grid in both horizontal and vertical directions.)

In contrast, the “scale-free-like” topography illustrated in Figure 5b is composed of a fragmented set of *islands of*
***A***, separated from each other by narrow channels of **B**. Two of the configuration variable values associated with this initial topography ($y_{2}$ and $z_{1}$) lie to either side of h=1.16. However, the value for $z_{3}$ lies far to the right on the graph of the analytic solution for the configuration variable values in terms of *h*.

A plausible reason for this divergence is that this “scale-free-like” system is human-designed, and has not yet beem brought to equilibrium. The extremely low value of $z_{3}=0.047$, which is far from the analytically-expected value of $z_{3}=0.075$ (if we were to nominate h=1.16), indicates that the actual presence of **A**-**B**-**A** triplets is much lower than we would expect if the system actually were at equilibrium centered on h=1.16. (See Table A3 in Section A.7 for details.)

A visual examination of Figure 5b shows that **A**-**B**-**A** triplets are indeed relatively rare. We would expect, if we bring this system to a CVM-based equilibrium point, that $z_{3}$ would migrate much more towards a value associated with h=1.16, i.e., we would expect $z_{3}\to 0.075$.

Expanding our view into the future, where we envision more detailed results where the activation energy $\varepsilon_{0}$ is allowed to vary as well as the interaction enthalpy $\varepsilon_{1}$ (or its corresponding *h-value*), it seems realistic that we can characterize a wide range of topographies using these two parameters.

Restricting the first-stage experiments, described in this work, to systems with equiprobable occurrences of nodes in **A** and **B** served two purposes. First, it made it possible to compare the actual configuration variable values achieved after free energy minimization with those predicted by the analytic solution. Second, it facilitated code verification and validation (V&V), described in Appendix A. Thus, for the experiments described here, the configurations shown in Figure 5 were kept relatively small, and had equiprobable numbers of nodes: both the (**a**) and (**b**) grids have 128 nodes each in state **A** and in state **B**.

The following Section 2.3 overviews how the 2-D CVM free energy equation is achieved. Understanding this equation will make it possible to understand the three key results presented in this paper, namely:Comparison of computational results with the analytic, where notable differences have been found,Identification of a useful parameter range, which allows us to identify the extent of the phase space that we will ultimately desire to map, andInitial topography characterization, which will allow us to correlate *h-values* with different kinds of topographies, as well as with their associated configuration variable values.

### 2.3. Result 1: Computational Configuration Variable Values Differ from Analytic

The original work by Kikuchi [1] and Kikuchi and Brush [2] used a zero-field Ising model, so that the enthalpy could be described with a single interaction enthalpy term. In this case (of zero-field, or a zero value for the activation enthalpy), an analytic solution is possible.

One of the primary goals of this work has been to computationally achieve systems that were at a free energy minimum, or equilibrium state, starting from various initial configurations (whether randomly-generated or designed to meet specific initial criteria). The intent was to assess the configuration variable values at these resulting equilibrium states, and to ascertain whether they indeed corresponded to the analytic solution. For this reason, the results reported in this work are confined to those where the zero-field condition is kept, so that the analytic solution should have been an accurate predictor of final results.

One of the most noteworthy computational results is that the *actual configuration variable values*, as computed for a free-energy-minimized grid at a given *h-value*, differ from the analytically *predicted values*. (The *h-value*, or simply *h*, is given as $h=exp(2\varepsilon_{1})$, and is an easier value for computational work than is $\varepsilon_{1}$.)

The 2-D CVM has an exact solution for the case where the activation enthalpy, $\varepsilon_{0}$, is zero. (That is, the solution exists when $x_{1}=x_{2}=0$, or the fraction of units in states **A** and **B** are equal.) This solution was initially presented, without details, in Kikuchi and Brush (1967) [2]. Maren (2019a) [4] gives the full derivation.

In order to compare computational results with the analytic, the computational experiments used topographies for which $\varepsilon_{0}=0$, or $x_{1}=x_{2}=0.5$.

As a point of origin, the basic Ising equation is
(1)$$\bar{F}=F/\left(Nk_{\beta }T\right)=\bar{H}-\bar{S},$$
where *F* is the free energy, *H* is the enthalpy and *S* is the entropy for the system, and where *N* is the total number of units in the system, $k_{\beta }$ is Boltzmann’s constant, and *T* is the temperature.

For working with abstract systems, the total $Nk_{\beta }T$ can be absorbed into a *reduced energy formalism*, as these values are constants during system operations. This leads to the *reduced representations* of $\bar{F}$, $\bar{H}$, and $\bar{S}$. We will work consistently with reduced representations throughout this work.

In a simple Ising model, both the enthalpy *H* and the entropy *S* can be computed based on only the relative fraction of active units in a bistate system. That is, there are only two kinds of computational units; *active* ones in state **A**, where the fraction of these units is denoted $x_{1}$, and *inactive* ones in state **B**, where the fraction of these units is denoted $x_{2}$. (Of course, $x_{1}+x_{2}=1.0$.)

In contrast to the simple entropy used in the basic Ising model in statistical mechanics, in the CVM approach, we expand the entropy term. The CVM entropy term considers not only the relative fractions of units in states **A** and **B**, but also those associated with the configuration variables, as described earlier in Section 2.1.

We can write the 2-D CVM free energy, using the formalism first introduced by Kikuchi in 1951 [1], and then further advanced by Kikuchi and Brush (1967) [2] (explicitly for the 2-D CVM), as
(2)$$\begin{array}{cc} {\bar{F}}_{2-D}=F_{2-D}/N= & \\ & {\bar{H}}_{2-D}-{\bar{S}}_{2-D}\\ & +\mu \left(1-\sum_{i=1}^{6}\gamma_{i}z_{i}\right)+4\lambda \left(z_{3}+z_{5}-z_{2}-z_{4}\right), \end{array}$$
where μ and λ are Lagrange multipliers, and we have set $k_{\beta }T=1$.

#### 2.3.1. The 2-D CVM Enthalpy

The enthalpy in a simple Ising system is traditionally given (using a mean-field approximation) as
(3)$$\begin{array}{cc} {\bar{H}}_{2-D}=H_{2-D}/N & \\ & ={\bar{H}}_{0}+{\bar{H}}_{1}\\ & =\varepsilon_{0}x_{1}+cx_{1}^{2}, \end{array}$$
where $\varepsilon_{0}$ and *c* are constants. (For a useful discussion of the mean-field approach as compared with the Bethe approximation, belief propagation, and the cluster variation method, see Yedidia [8].)

The first term on the RHS (Right-Hand-Side) corresponds to the *activation enthalpy*, or enthalpy associated with each active unit. The second term on the RHS corresponds to the *interaction enthalpy*, or energy associated with pairwise interactions between active units.

In contrast, the enthalpy for the 2-D CVM is given as
(4)$$\begin{array}{cc} {\bar{H}}_{2-D}=H_{2-D}/N & \\ & ={\bar{H}}_{0}+{\bar{H}}_{1}\\ & =\varepsilon_{0}x_{1}+\varepsilon_{1}\left(-z_{1}+z_{3}+z_{4}-z_{6}\right) \end{array}$$

Note that in the original work by Kikuchi and Brush, Equation (Equation 4) is simplified (K&B Equations (I.16) and (I.17)) to
(5)$${\bar{H}}_{2-D}=\varepsilon_{1}\left(-z_{1}+z_{3}+z_{4}-z_{6}\right),$$
that is, they omit the term linear in $x_{1}$; the activation enthalpy. (This means that they are working with a zero-field system.)

We can infer that Kikuchi and Brush omit the activation enthalpy from their enthalpy term because they move directly to the analytic solution for the 2-D CVM free energy, which is solvable only in the equiprobable distribution case of $x_{1}=x_{2}=0.5$. The equiprobable distribution is achieved only when the activation enthalpy is zero; that is, when $\varepsilon_{0}=0$.

When the activation enthalpy $\varepsilon_{0}>0$, then the units in state **A** have an energy associated with them that is greater than that of the units in state **B**. Thus, an equilibrium solution will favor having fewer units in state **A**. There is no analytic solution for this case, other than that in which the interaction enthalpy is zero ($\varepsilon_{1}=0$). This latter case is trivial to solve, and is not particularly interesting, as the distribution of different kinds of nearest-neighbor pairs and triplets will be probabilistically random, excepting only that the proportions of units in states **A** and **B**, respectively, will be skewed by the activation enthalpy parameter $\varepsilon_{0}$.

As just noted, the typical expression for the interaction enthalpy is a quadratic term in $x_{1}$, that is, $H_{1}=cx_{1}^{2}$. The parameter *c* encompasses both the actual interaction energy for each pairwise interaction, and a constant that expresses the distribution of pairwise interactions as a simple linear function of the fraction of active units ($x_{1}$) surrounding a given active unit. This is then multiplied by the total fraction of active units, giving the quadratic expression.

In the expression for the 2-D CVM interaction enthalpy, we have terms that expressly identify the total fraction of nearest-neighbor “unlike” pairs ($y_{2}$) and “like” pairs ($y_{1}$ and $y_{3}$). Thus, we can replace $cx_{1}^{2}$ with the fraction of “unlike“ pairs (counted twice, to account for the degeneracy in how these pairs can be counted), and the fractions of “like” nearest neighbor pairs.

We recognize that this is a simplification; we are not counting interaction energies due to next-nearest neighbor pairs, the $w_{i}$, nor from the triplets $z_{i}$. We are, effectively, subsuming these into the pairwise interactions that are being modeled with the $y_{i}$.

We take the interaction enthalpy parameter $\varepsilon_{1}$ to be a positive constant.

We envision a system in which the free energy is reduced by creating nearest-neighbors of like units, that is, **A**-**A** or **B**-**B** pairs. (Decreasing the interaction enthalpy leads to decreasing the free energy, which is desired as we go to a free energy minimum, or equilibrium state.) Similarly, the interaction enthalpy should increase with unlike pairs, or **A**-**B** pairs (or vice versa).

Thus, we write the interaction enthalpy equation, following the formalism introduced by Kikuchi and Brush, as
(6)$$\begin{array}{c} {\bar{H}}_{2-D}=H_{2-D}/N=\varepsilon_{1}\left(2y_{2}-y_{1}-y_{3}\right). \end{array}$$

We interpret this equation by noting that as we increase the fraction of unlike unit pairings ($2y_{2}$, encompassing both **A**-**B** and **B**-**A** pairs), we raise the interaction enthalpy. At the same time, if we’re increasing *unlike* unit pairings, we are also decreasing *like* unit pairings (**A**-**A** and **B**-**B**, or $y_{1}$ and $y_{3}$, respectively), so that we are again increasing the overall interaction energy. (Note that the *like* unit pairings show up in Equation (Equation 6) with a negative sign in front of them.)

Thus, if we create a 2-D CVM that is like a checkerboard grid (alternating unit types), then we are moving to a higher interaction enthalpy and higher free energy, and away from a free energy minimum. If we create a grid such as that shown in the LHS of Figure 5, with all of the units in each state grouped together, respectively (the “extreme rich club-like” pattern), then we lower the interaction enthalpy and correspondingly lower the free energy.

Clearly, if the interaction enthalpy were all that mattered, we would have a 2-D CVM pattern such as the one on the left in Figure 5. What keeps this from happening is the entropy term, which demands some distribution over different possible configurations.

Before moving on to the entropy, we note that we can express the interaction enthalpy using the triplet configuration variables $z_{i}$, instead of the nearest-neighbor pair variables $y_{i}$. We can do this by drawing on *equivalence relations* between the $y_{i}$ and $z_{i}$ variables. Those for $y_{2}$ are given (Kikuchi and Brush (1967) [2], and also Maren (2019a) [4]) as
(7)$$y_{2}=z_{2}+z_{4}=z_{3}+z_{5}$$
(8)$$2y_{2}=z_{2}+z_{4}+z_{3}+z_{5}.$$

Notice that we have two ways of expressing $y_{2}$ in terms of the $z_{i}$, as shown in Equation (Equation 7). Since we want to work with the total $2y_{2}$, it is easy to express that as the sum of the two different equivalence expressions. This proves useful when deriving the analytical solution for the free energy minimum, or equilibrium point.

We also have equivalence relations for $y_{1}$ and $y_{3}$, given as
(9)$$y_{1}=z_{1}+z_{2}$$
and
(10)$$y_{3}=z_{5}+z_{6}.$$

This lets us write
(11)$$\begin{array}{c} {\bar{H}}_{2-D}=\varepsilon_{1}\left(2y_{2}-y_{1}-y_{3}\right)=\varepsilon_{1}\left(z_{4}+z_{3}-z_{1}-z_{6}\right). \end{array}$$

As a minor note, the enthalpy used in previous related work by Maren [3,20], was
(12)$$\begin{array}{c} {\bar{H}}_{2-D}=H_{2-D}/N=\varepsilon_{1}\left(2y_{2}\right)=\varepsilon_{1}\left(z_{2}+z_{3}+z_{4}+z_{5}\right). \end{array}$$

The results given here are similar in form to the results presented in the two previous works by Maren; they differ in the scaling of the interaction enthalpy term.

#### 2.3.2. The 2-D CVM Entropy

The entropy for the 2-D CVM is given as
(13)$$\begin{array}{cc} {\bar{S}}_{2-D}=S_{2-D}/N= & \\ & 2\sum_{i=1}^{3}\beta_{i}Lf\left(y_{i}\right))+\sum_{i=1}^{3}\beta_{i}Lf\left(w_{i}\right)\\ & -\sum_{i=1}^{2}\beta_{i}Lf\left(x_{i}\right)-2\sum_{i=1}^{6}\gamma_{i}Lf\left(z_{i}\right), \end{array}$$
where Lf(v)=vln(v)−v.

A more detailed discussion of the entropy term is given (Kikuchi and Brush (1967) [2] and also Maren (2019a) [4]).

#### 2.3.3. Free Energy Analytic Solution

Kikuchi and Brush (1967) [2] provided the results of an analytic solution for the 2-D CVM free energy, for the specific case where $x_{1}=x_{2}=0.5$. Specifically, making certain assumptions about the Lagrange multipliers shown in Equation (Equation 2), we can then express each of the configuration variables in terms of $\varepsilon_{1}$.

More usefully, since the expression actually involves the term $exp(2\varepsilon_{1})$, and not $\varepsilon_{1}$ itself, it is much easier to use the substitution variable $h=exp(2\varepsilon_{1})$. We refer to *h* (or sometimes, the *h-value*), as the *interaction enthalpy parameter* throughout.

The full derivation of the set of equations giving the configuration variable values at equilibrium (i.e., at $x_{1}=x_{2}=0.5$) is given in Appendix A of Maren (2019a) [4]).

The full set of the analytic expressions for the various configuration variables is couched in terms of a denominator involving *h*, specifically
(14)$$\Delta =-h^{2}+6h-1,$$
which Kikuchi and Brush present as their Equation (I.24) [2].

We recall that at the equiprobable distribution point, where $x_{1}=x_{2}$, we have a number of other equivalence relations, e.g., $z_{1}=z_{6}$, etc.

We then (following Kikuchi and Brush, in their Equation (I.25)) identify each of the remaining configuration variables as
(15)$$\begin{array}{lll} y_{1}=y_{3} & = & \frac{\left(3h-1\right)}{2\Delta }\\ y_{2} & = & \frac{h(-h+3)}{2\Delta }\\ w_{1}=w_{3} & = & \frac{{\left(h+1\right)}^{2}}{4\Delta }\\ w_{2} & = & \frac{\left(3h-1)(-h+3\right)}{4\Delta }\\ z_{1}=z_{6} & = & \frac{\left(3h-1)(h+1\right)}{8\Delta }\\ z_{2}=z_{5} & = & \frac{\left(3h-1)(-h+3\right)}{8\Delta }\\ z_{3}=z_{4} & = & \frac{\left(-h+3)(h+1\right)}{8\Delta } \end{array}$$

#### 2.3.4. Divergence in the Analytic Solution

As is obvious from Equations (Equation 14) and (Equation 15), there will be a divergence in the analytic solution for the configuration variables when Δ=0, because the analytic solution contains a denominator term (a scalar factor of Δ) that is quadratic in *h*. Specifically, the term diverges for h=0.172 or h=5.828.

We are interested in the latter case, where the value of h>1 indicates that $\varepsilon_{1}>0$, which is the case where the interaction enthalpy favors like-near-like interactions, or some degree of gathering of similar units into clusters. This means that we expect that the computational results would differ from the analytic as h→5.828, or equivalently, for $\varepsilon_{1}\to 0.8813$ where $h=exp(2\varepsilon_{1})$. (Actually, the computational results diverge from the analytic substantially before this point.)

This naturally has some impact, but we find that the real point of interest comes from examining the impact of higher *h-values* on the free energy itself. Increasing *h* increases the overall value of the enthalpy term, and makes it overwhelmingly dominant with regard to the entropy term.

#### 2.3.5. Significance of *h* in the Free Energy

Because it is the *minimum in the negative entropy* that creates the possibility for minimizing the free energy, we need to have the entropy term play a role that is at least on par with the enthalpy. If the enthalpy term is overall too large, then there will be an adverse effect on finding a useful free energy minimum.

The impact of the *interaction enthalpy parameter h* on the overall free energy is shown in Figure 7, which presents the results when $x_{1}=0.5$, which is the case where all of the results should conform with the analytic solution.

### 2.4. Result 2: Identification of a Useful Parameter Range

One of the important precursor steps is to identify how the thermodynamic quantities behave, for a system brought to a free energy minimum for various *h-values*. To address this, and with no attention paid to topographies (which have truly been the focus of this work), a series of experiments were conducted with randomly-generated initial patterns.

Figure 7 shows that beyond a certain value, approximately h>1.2, the enthalpy term dominates the entropy, and thus further increasing *h* does not yield an improved model. Further, as we are primarily interested in ferromagnetic systems (where *like* nodes tend to cluster with *like*), we are not interested in the antiferromagnetic regime, where h≤1.0 ($\varepsilon_{1}\leq 0$.). This work was done using a set of codes that brought a randomly-generated system to a minium for different *h-values*, and then gently “perturbed” the result, and allowed it to come to free energy again. The “perturbation” code is available in the author’s GitHub directory; see further details in Section A.4. All code is publicly available. See *Sample Availability*.)

Figure 8 identifies the reasonable phase space region for mapping out the configuration variable values for at-equilibrium patterns as a function of the activation and interaction enthalpies ($\varepsilon_{0},\varepsilon_{1}$). The useful region is bounded by the area where $\varepsilon_{0}<3.0$ and $\varepsilon_{1}<0.25$ (h<1.649).

#### 2.4.1. Experimental Trial 1.a: Very Large *h-Value*

As a very preliminary investigation, the “extreme rich club-like” pattern shown in Figure 5a was brought to free energy equilibrium for three trials where h=2.0. This meant that we were knowingly using an *h-value* that was larger than was reasonable, based on the thermodynamic values shown in Figure 7. The objective was to bring the system to equilibrium at an *h-value* that would force the extreme clustering shown in Figure 5a, making the resulting system very close to the initial pattern. Even for this (relatively extreme) *h-value*, the equilibrium state moved substantially away from the original, introducing much more varied topographies. Details are presented in Section A.5.1; see also *Sample Availability*, which includes experimental documentation.

#### 2.4.2. Experimental Trial 1.b: Two Patterns at Same Large *h-Value*

One of the most important questions that could be asked was: how would two entirely different topologies respond, if they were brought towards an equilibrium state for the same *h-value*?

To address this, the two initial grid patterns used in this work, the “extreme rich club-like” and the “scale-free-like” patterns were each brought towards an equilibrium state for h=1.65. This *h-value* was selected because it seemed to be central to the initial configuration variable values for the “extreme rich club-like” pattern. Thus, the expectation was that the “extreme rich club-like” topography would not change much, but that the “scale-free-like” topography would move towards a state where the associated configuration variable values were more aligned with those around h=1.65.

The results were surprising.

The following Figure 9 shows both the initial and final configuration variable values $z_{1}$, $z_{3}$, and $y_{2}$, for both the “extreme rich club-like” and the “scale-free-like” patterns, brought towards equilibrium (with the limit of maximally using 100 swaps), for h=1.65. The initial values for each are shown with diamonds marked with ***i***. (These were initially presented in Figure 6.) The resulting values for the “extreme rich club-like” pattern are denoted with **a**, and those for the “scale-free-like” pattern are denoted with **b**.

The following Table 2 gives selected configuration variable values corresponding to the two patterns used in this experiment; the “scale free-like” and the “extreme rich club-like” patterns, both before and after (moving towards) an equilibrium at h=1.65. The objective was to see how far, with a limit of 100 swaps, the two patterns could each be brought close to the analytically-predicted configuration variable values. (See Section 4.2 for details; see also *Sample Availability*, which includes experimental documentation.)

Somewhat surprisingly, even for such an extreme *h-value* of h=1.65, the “extreme rich club-like” pattern changed topography significantly (see Section 2.5). That is, it did not stay with the configuration variable values indicated by the analytic soluton associated with h=1.65. Instead, it moved towards far greater topographic complexity than was originally envisioned.

The “scale-free-like” pattern did not reach a final at-equilibrium state before reaching the stopping limit of 100 swaps, but did move in the expected direction, for $y_{2}$ and $z_{3}$. (The result for $z_{1}$ did not get as far. See a comparison of the final states for the “scale-free-like” pattern with two different *h-values* in the following Section 2.5 for further discussion.) The reason that the resulting pattern did not move further towards the configuration variables indicated at h=1.65 (particularly for $z_{3}$) is most likely due to the limitations of the “swap-and-test” nature of the free energy minimization algorithm. The run was still selected for presentation because it exemplified the kinds of patterns that were evolving over time.

Also, given the behavior of the “extreme rich club-like” pattern when h=1.65, it is clear that the at-equilibrium configuration variable values associated with any of the higher *h-values* will need to be identified via multiple experimental trials, likely beginning with a range of initial topographies.

The actual topographies that resulted are shown in Figure 10. The resulting pattern for the “extreme rich club-like” topography, shown in Figure 10a, is interesting in that the pattern that evolved, from the original highly-massed pattern (shown in Figure 5a), shows not only the expected “islands” that formed, as random pairs of nodes were selected and switched, but also two “peninsulas” that emerged as inroads into the original “landmasses.”

The resulting pattern for the “scale-free-like” topography, shown in Figure 10b, is dominated by extended “legs” with widths of one, two, or three nodes, and interconnections between these peninsula-like legs, as well as several remaining “islands” from the original topography. The boundaries in this figure are irregular. Finding a pair of nodes where switching their respective states would significantly reduce the free energy, using a random node-pair selection, is not likely to be fruitful.

In order to bring various topographies closer to equilibrium, faster, it will be necessary to devise algorithms that are sensitive to the existing topographies. This will require an object-oriented approach, and is part of an ongoing investigation. Some of these research goals for the next stage of work are presented in Section 3.

In summary, even for an *h-value* that is very high (h=1.65), bringing a system close to an equilibrium state produces results that introduce a greater distribution across the various configuration variables, rather than allowing tight clustering. Setting the *h-value* to 1.65 is at the outer edge of a useful parameter setting, according to the free energy computations shown in Figure 6 and Figure 7. This is obvious for the case where the initial pattern is tightly clustered, as shown in Figure 5a. In this case, the value for $z_{1}$ (representing **A**-**A**-**A** triplets) decreases, which means that clustering of *like-with-like* decreases.

In the case of the initial “scale-free-like” pattern, as was the case for the initial pattern in Figure 5b, free energy minimization at a high *h-value* also decreased the *like-near-like* connections; i.e., decreased $z_{1}$. The likely reason is that the initial pattern was, itself, so far from equilibrium that as the system moved towards an equilibrium state, a large fraction of the swaps reduced overall free energy, even if not associated with that particular *h-value*. This suggests that more fruitful results will be obtained when initial patterns are already somewhat closer to an equlibrium state. Work with this “scale-free-like” pattern is discussed more fully in the following Section 2.5.

### 2.5. Result 3: Topography Characterization

The early experiments with a 2-D CVM system were all done using various patterns of 256 (16 × 16) nodes (units) each. Two of these patterns were illustrated previously in Figure 5; these were (**a**) the “extreme rich club-like” pattern and (**b**) the “scale-free-like” pattern.

Keeping initial experiments confined to this relatively small 16 × 16 (256-node) grid allowed three things:Sufficient variety in local patterns–this grid size was large enough to illustrate several distinct kinds of topographies (each corresponding to different *h-values*),Sufficient nodes–there were enough nodes so that triplet-configuration extrema could be explored in some detail, e.g., for relatively small numbers of nodes in state **A** (results not reported in this work), andCountability–the verification and validation (V&V) effort required that several early versions of the grid be *manually counted* for all the configuration values for a given 2-D grid configuration, and matched against the results from the program.

One final advantage of the 16 × 16 grid layout was that the different grid configurations were both large enough to show diversity, but small enough so that it was possible to manually create figures illustrating the activation states (**A** or **B**) of each node, thus illustrating the detailed particulars of each configuration design. (The code at this point included only rudimentary print-outs of grid patterns.)

The experiments began with manually-designed pattern configurations, such as the two shown in Figure 5. These two configurations correspond (approximately) to the notions of “scale-free” and “rich club” topographies. (For more details on how the patterns were designed, with particular attention to the “scale-free-like” pattern addressed in this subsection, see Section A.6.)

Experimental results for Trial 1, performing free energy minimization for the “extreme rich club-like” pattern (see Figure 5a) were given previously in Section 2.4.2. Thus, this subsection addresses how the experiments were conducted for Trial 2, working with the “scale-free-like” pattern shown in Figure 5b.

The configuration on the right of Figure 5 (pattern **b**) is an effort to build a “scale-free-like” system. The notion of a “scale-free” system is that the same kind of pattern replicates itself throughout various scales of observation in a system. More specifically, following an interpretation offered by Jiang and Yin (2014) for fractal systems, there are “far more small things than large ones” [19].

After the pattern was designed, the configuration variable values were computed. The details for $z_{1}$ and $z_{3}$ are shown in Figure 11.

As with the experiments done on the “extreme rich club-like” pattern discussed in the previous subsection, we began with the “scale-free-like” initial grid, introduced earlier in Figure 5b, and shown in more detail in Figure 11. This initial grid was brought to (or towards) a free energy minimum point, for each of two different *h-values*:Trial (2.a): h=1.65, which was expected to push the system substantially towards increased like-near-like clustering, andTrial (2.b): *h* = 1.16, which was expected would *not* dramatically change the pattern configuration, as the current configuration values indicate that an *h-value* of 1.16 is approximately close to its current state. (See Figure 6; the initial configuration values for the “scale-free-like” pattern approximately surround *h* = 1.16.)

The following Figure 12 illustrates the results described in the following three subsections. Using the “scale-free-like” pattern as a starting point, this figure depicts the illustrative configuration variable values $z_{1}$, $z_{3}$, and $y_{2}$, for the initial grid pattern (where the light-colored diamonds marking the configuration variable values are denoted with “i”), and for the systems brought towards (but not necesarily reaching) equilibrium states, where the diamonds are denoted “**a**” (for *h* = 1.65) and “**b**” (for *h* = 1.16).

#### 2.5.1. The Configuration Variable Values for the Initial Scale-Free-Like Pattern

The configuration variable values for the initial “scale-free-like” pattern show that it is substantially far from equilibrium, for any possible *h-value*. Specifically, the value for $z_{3}=0.0469$ (furthest diamond to the right on the $z_{3}$ graph in Figure 12) indicates a very low fraction of triplets in the **A**-**B**-**A** configuration. When we refer to the original pattern design, e.g., Figure 11, we see that the initial design favors masses of units in state **A**, separated from one another by one, two, or three units in state **B**. The instances where two masses of **A** units are separated from one another by only a single-unit channel of **B** units are relatively rare. In contrast, looking ahead to the results in Figure 13, we see more instances of extended peninsulas of **A** units, separated from each other by narrow rivulets of **B** units. (For details on the actual configuration variable values, both in the initial state and after being brought closer to equilibrium, see Section A.7.)

#### 2.5.2. Experimental Trial 2.a: “Scale-Free-Like” Grid with *h* = 1.65

We limited the number of total “swaps” allowed in the system to 100, as an effort to bring the system from Figure 11
***in the direction of*** an at-equilibrium state h=1.65. After 100 swaps, we get the result shown in Figure 13a. In contrast, the pattern on the right, Figure 13b, illustrates a topography for the same initial pattern, but brought (closer) to a free energy minimum for h=1.16. (See details in Table A3.)

With such a limited number of swaps, there was no intention of actually reaching the desired equilibrium state for the case where h=1.65. Rather, the intent was to identify how the topography would change when the *h-value* indicated a greater degree of like-near-like clustering.

We see in Figure 13a that, as expected, the clusters are coalesced as compared with the original pattern. As is clear from results reported in the prior two subsections, h=1.65 is very large compared with what is a reasonable target for this pattern. (See Figure 6) This trial was conducted as an initial exploration of how a starting pattern would change when the selected *h-value* would be likely to drive the system far from its original topography.

#### 2.5.3. Experimental Trial 2.b: “Scale-Free-Like” Grid with *h* = 1.16

When we bring the system from Figure 11 (closer) to free energy equilibrium with h=1.16, we get the result shown in Figure 13b.

With a lower *h-value* than was previously used, we expect that the resulting (close to an) at-equlibrium state will indeed have more *like-with-like* unit pairs than there would be if there were no interaction enthalpy, but fewer than if the *h-value* were higher, as was used with h=1.65 in the previously-discussed Trial 2.a. What we see is interesting. It is not surprising, but it is worth noting that we get a number of “spider leg” connections between what remains from the original clusters.

These “spider legs” give us *both like-near-like* (extending the length of the spider leg) as well as *like-near-unlike* (on either side of the spider leg) connections for each unit in the leg. Thus, each unit in a spider leg gives us approximately the ratio of $y_{1}$ and $y_{2}$ pairwise connections (two $y_{2}$ connections per $y_{1}$, accounting for the factor of two degeneracy in the $y_{2}$ pairs) that we’d expect near h=1.0.

Overall, for both Trials 2.a and 2.b, we see that the $z_{3}$ value has been brought much closer to what would be expected when h=1.16 (and further from what would be expected from h=1.65). In fact, for Trial 2.b, the resulting $z_{3}$ value is very close to what we would expect for h=1.16.

Changing the $z_{3}$ values within the topography is not easy when using the simple random-swap approach employed in the current algorithm. It is likely that a more advanced, topographically-sensitive algorithm would produce better results, faster.

It is likely that much of the effort moving towards an at-equlibrium state was dictated by adjusting the $z_{3}$ values, as the initial $z_{3}$ value here was so far from what would have been expected from the *h-values* indicated by the other configuration values.

There was little movement in the $y_{2}$ values, although we note that the resulting $y_{2}$ moved in the desired direction for the case where h=1.65 and in the other direction for h=1.16. At this stage, we cannot make more detailed comments, as the movement towards an equilibrium state is subject to random choices for node swapping, with consequently non-predictable results.

Finally, the changes in $z_{1}$
*both* move contrary to the desired directions; this is likely an artifact of bringing the pattern with such an extreme initial value for $z_{3}$ into closer alignment with an equlibrium-state configuration.

What we *can* infer, through visual examination of these two topographic instances, is the formation of the previously-mentioned “spider legs” connecting various previously unconnected masses. If we refer back to Figure 10, we see that the original single, large mass of **A** units was broken up and similar “spider legs” of **B** in the midst of **A** appeared, as well as shorter “spider legs” connecting what might have been initially isolated units of **A** that could have emerged early in the move towards an equilibrium state.

In summary, we are seeing some emerging topographic elements that may prove to be characteristic of 2-D CVM grid patterns that are brought to equilibrium, for a range of *h-values*.

## 3. Discussion

The results presented in the preceding Section 2 are the first steps in what will potentially reveal an interesting means for characterizing two dimensional topographies using the 2-D CVM. To sketch future directions, this Discussion addresses three topics:Next steps in characterizing 2-D CVM topographies and their associated configuration variable and $\left(\varepsilon_{0},\varepsilon_{1}\right)$ parameter space values, as well as strategies to move from one topography to another,Potential role of using the 2-D CVM to model certain structural aspects of brain organization, and also addressing the role of free energy minimization within the brain, andNext steps to connect 2-D CVM topographies with existing models and characteristics for different forms of structural organization.

### 3.1. Next Steps in Developing 2-D CVM Topographies and the Associated Parameter Space

The approach taken here, albeit somewhat brute-force, reveals interesting topographies that suggest future steps. Among the most pertinent next steps are the following:Map the set of configuration variables values to the $\left(\varepsilon_{0},\varepsilon_{1}\right)$ parameter space (noting that there may be a configuration variable value range, depending on the initial topography),Characterize how visual topography features correspond with configuration variable sets, and devise algorithms to characterize topographic elements (e.g., clusters) in terms of variables such as their relative size distributions, compactness, and other measures which are typical in other 2-D grid systems, noting that the algorithms need to be created specific to the grid layout for the 2-D CVM described here, andDevise strategies to efficiently transit from one topography to another, as there are changes in the parameters $\left(\varepsilon_{0},\varepsilon_{1}\right)$.

As a first step, we need to fill out the phase space.

In this work, two plausible data points for configuration variable values were identified for positions along the left-hand-side axis, where $\varepsilon_{0}=0$. These were when h=1.16 and h=1.65. Values along the top horizontal axis are easy to compute (see Maren (2019a) [4]).

It will be somewhat more challenging to compute configuration variable values for the interior of this phase space. The reason is that, even though we have found that the computated values for the configuration variables diverge from the analytic as $\varepsilon_{1}$ moves further away from zero, the analytic solution is still a useful reference point, especially in the neighborhood of $\varepsilon_{0}\approx 0$. Finding interior points will require a greater number of trials per candidate set of ($\varepsilon_{0},\varepsilon_{1}$) values. In particular, until a good object-oriented method, using topographic information, has been devised, it is likely that best results will be achieved by creating initial “pretty good” topographic patterns, and then bringing them to equilibrium. This will require a first round of experimentation to determine what these initial patterns should be.

Once a set of configuration variable values have been established for a a suitable set of ($\varepsilon_{0},\varepsilon_{1}$) values, it will likely be possible to establish gradients throughout the phase space and produce approximate sets of configuration variables for any position within the ($\varepsilon_{0},\varepsilon_{1}$) phase space.

Work by Pelizzola (2005) on the CVM showed that belief propagation methods could be used to bring the system to a free energy minimum [9]. While message-passing algorithms are important, the way in which the 2-D CVM has been approached here emphasizes the topographic nature of the free energy-minimized systems.

Figure 10 showed two different topographies, each for the same enthalpy parameter set, where the interaction enthalpy $\varepsilon_{0}=0$ and the interaction enthalpy $\varepsilon_{1}=0.250$ (*h-value* = 1.65). The topography for Figure 10a reached a free energy equilbrium before the limit of 100 trial swaps was completed; the topography in Figure 10b did not.

The topography in Figure 10a resulted from an “extreme rich club-like” starting point. The *h-value* selected for the free energy minimization was 1.65, which is relatively high-and was reasonably expected to correspond to system with extreme like-near-like clustering. (See Figure 6 and Figure 8, and also Figure 7 to see that an *h-value* of 1.65 is actually a bit too extreme; it was selected to push the envelope for the extreme case of a highly like-near-like system.)

The results from the initial experiments with high *h-values* suggest that near-term efforts should include creating initial topographies that would potentially be closer to at-equilibrium states for different enthalpy parameter combinations, and to iterate that process until a robust set of at-equilibrium topographies and configuration variable values have been achieved for a given pair of enthalpy parameters.

We can envision being able to characterize various topographic *regions* within the two-parameter phase space used for the 2-D CVM. The process of filling in the actual phase space details, correlating configuration variable values (and their associated thermodynamic quantities) with $\left(\varepsilon_{0},\varepsilon_{1}\right)$ pairs, will be relatively straightforward. It will, of course, be greatly aided by developing the next generation of code, preferably written in an object-oriented manner.

With the phase space mapped out, several enticing avenues for future work open up. Among the first tasks are:Devising a means to correlate phase space regions with various topograhic descriptions, e.g., “rich club,” andDevising strategies to efficiently transit from one topography to another, as there are changes in the parameters $\left(\varepsilon_{0},\varepsilon_{1}\right)$.

These steps, however important, simply pave the way for the next generation of work.

One of the most interesting notions advanced by Yedidia, Weiss, and Freeman (2002) has been that the CVM method was a means to accomplish belief propagation [6]. Belief propagation, as we know, is essentially a symbolic-level activity, initially proposed by Pearl [21]. The belief propagation method is intrinsically associated with graphical models [10]. The challenge is that, in a traditional belief-propagation approach, each node in a graph is associated with a specific belief. In a CVM-based approach, a cluster or pattern of nodes might evidence in correspondence with a specific feature. Thus, we are interested not only in correlating a distributed representation with a more classically-symbolic one, but also in identifyng how dynamic patterns in one can correlate with emerging beliefs.

Another interesting avenue for work will be the connection between modeling long-range order in the brain and in other (non-biological) systems with the connectivity between clusters of active nodes that emerge following 2-D CVM free energy minimization. As an example, the topographies resulting from free energy minimization on the scale-free-like initial pattern (in Figure 13) showed *spider leg-like* connecting branches between what had previously been isolated masses. These spider legs, running diagonally across the grid, suggest something like long-range order. Clearly, many more experiments are needed to validate the exact conditions under which these spider leg extensions appear.

Following initial investigations into how topologies in 2-D CVM grids may show apparent long-range connection paths between different activation clusters, it would be potentially fruitful to make some connections between these empirical results and other studies involving long-range order. As an interesting corollary to long-range order in neural systems, non-biological systems with repulsive long-range connections can exhibit long-range order in the form of stripes [22], using a combination of short-range and long-range interactions. It would be interesting, and potentially useful, to investigate the introduction of long-range interactions into a 2-D CVM with the entropy.

As an evolution from studies on long-range order, the role of free energy minimization in a 2-D CVM grid can also potentially apply to message passing. Raymond and Ricci-Tersenghi (2013) report on message passing as a means for correcting beliefs [23]. More recently, Parr et al. discuss neuronal message passing [24], with a good review of relevant literature. Recent work on neural connectivity in the brain by Peraza-Goicolea et al. (2020) addresses susceptibility propagation (SP), belief propagation (BP), and critical state dynamics [25], and Friston, Parr, and de Vries (2017) investigate the connection between belief propagation and active inference [26]. Thus, the substantial work on message passing in neural systems may usefully inform further investigations on using a 2-D CVM as a means for modeling neural systems.

The following subsection briefly describes prior work on free energy minimization in the brain, and the next subsection describes different topographies (“rich club,” “small-world,” and “scale-free”), both in general and as they occur in the brain.

### 3.2. Free Energy Minimization in the Brain

The notion that the brain operates to minimize free energy is not new; there is a long and robust history of efforts to advance this approach. The intent of this subsection is not to provide a comprehensive review, but to identify key works in the evolution of thought about the role of free energy in neural systems.

Kozma et al. (2004, and see also references cited therein) advanced the notion of *neuropercolation*, or percolation theory as applied to neural systems [27]. These early studies often focused on random cellular automata (RCA) on a lattice, which led to identification of critical phenomena and scaling properties. Their results “… indicate that brains maintain themselves at the edge of global instability by inducing a multitude of small and large adjustments in the form of phase transitions. Phase transitions mean that each adjustment is a sudden and irreversible change in the state of a neural population.”

About concurrently, and often in conjunction with Kozma, Freeman et al. (2006) was developing the notion of *metastable states* in the brain [28].

Further early work includes notions of criticality in the brain induced by “mutual excitation,” as suggested by Kozma et al. (2012) [29] and Kozma and Puljic (2015) [30]. This work suggests that the brain operates at a near-critical state, specifically that the brain operates at a “pseudo-equilibrium” [29]. Building on the notion of *neuropercolation*, they suggest that “populations of cortical neurons … sustain their metastable state by mutual excitation, and that its stability is guaranteed by the neural refractory periods.”

Others, such as Tognoli and Kelso (2014), also suggest that the brain operates in a metastable state [31]. Wilting and Priesemann (2020) provide an excellent review and discussion of criticality in neural systems [32].

Recent data-driven studies continue to support the notion that the brain acts near criticality. Ezaki et al. (2020) have used functional magnetic resonance imaging (fMRI) of networks at a whole-brain level to posit that individuals with higher fluid intelligence have brains operating closer to criticality, as compared with others [33]. They address patterns of transition between discrete states.

Beginning in the second decade of this millenium, Friston and colleagues began introducing the notion of free energy minimization as a key function in brain processes (2010, 2013, 2015, 2020) [34,35,36,37]. One of the fundamental ideas that they have advocated (Friston et al., 2012) shows how Bayesian priors can be derived as outcomes of variational free energy minimization [38]. This specifically addresses notions of self-organization within the brain.

As noted by Friston et al. (2013), movement towards a new state will be influenced by the nature of the stimulus or perturbation. Friston describes this, saying: “In short, the internal states will appear to engage in Bayesian inference, effectively inferring the (external) causes of sensory states. Furthermore, the active states are complicit in this inference, sampling sensory states that maximize model evidence: in other words, selecting sensations that the system expects. This is active inference, …” [35].

Friston and colleagues have taken this further to suggest mechanisms for neural communication strategies as an operative element of a “graphical brain” [26]. More recently, Demekas, Parr, and Friston (2020) have suggested that free energy minimization in the brain is a principle underlying active inference, and can facilitate emotion recognition [39].

The arguments advanced by Friston and colleagues are not without some controversy. A recent publication by Biehl et al. (2020) suggests counterarguments [40]. One of the most useful means of resolving these questions may be to formulate systems with Markov partitions at various scales, and to simulate behaviors that would emerge from such systems.

Friston et al. (2014), using resting state fMRI timeseries to describe patterns or modes of distributed activity underlying brain functional connectivity, also show the connection to effective connectivity. They conjecture that dynamic instability is inevitable in a system separated from external states by a Markov blanket, and further suggest that this leads to a free energy formulation for nonequlibrium steady-state dynamics [41]. Friston and colleagues carry this line of thinking forward (2020) by addressing how a recursive application of parcellation of states into Markov blankets allows dynamic expresion at increasing scales [42]. It is premature to envision an actual correspondence between topographies as produced by CVM free energy minimization and recursive Markov blanket partitions, however, this is a tantalizing prospect for future work as means of characterizing 2-D CVM topographies evolve.

Maren (2019b) [43] has illustrated how the 2D CVM could play a role within a computational engine modeled on notions advanced by Friston and others for active inference; a topic that can be investigated once the avenues identified in this section have been developed to sufficient maturity.

A detailed investigation into these different points of view is beyond the scope of this work. It suffices to say that various leading researchers are proponents for the idea that the brain does, indeed, perform some sort of free energy minimization.

Once we have mapped correlations between visual topographic characterization, configuration variables, and enthalpy parameters (as described in the preceding subsection), there are several intriguing avenues for further investigation. The most interesting of these rely on characterizing dynamics, that is, evolving pathways through the enthalpy parameter phase space. These avenues include:Investigating means to “reverse engineer” empirical timeseries (for example, natural topography evolution) via identifying an enthalpy phase space trajectory, andInvestigating the potential for dynamic graph generation, via a trajectory through the enthalpy phase space; this could potentially be applied to Bayesian model selection and/or message-passing.

These avenues will be discussed in further works, especially as the impact of phase space trajectories becomes more well-characterized.

### 3.3. The Challenge: Efficiently Characterizing Two-Dimensional Topographies

The following Figure 14 illustrates a set of three easily-observed, naturally-occurring topographies; black lava rocks surrounded by white coral sand. Figure 14a shows relatively compact black lava rock elements, all of which can be connected with each other, although some connection routes will pass through neighboring masses. In Figure 14b, there is a greater range of sizes in the black lava rock masses. Further, some are now disjunct from each other; they are now truly “islands.” In Figure 14c, the trend towards greater granularity continues, as the lava rocks are broken into smaller and smaller units, and relatively more are isolated from each other.

Just as Figure 14 illustrates a continuum of topographies, a potentially fruitful line of work will be to address how 2-D CVM networks correspond to previously-identified topography types, whether they are extant in nature or are more abstract. Three common topographic types have been recognized with their own labels, specifically:Rich club,Small-world, andScale-free.

Ultimately, we desire a means of connecting the morphological descriptions and associated topographic or clustering measures with various topology patterns produced via free energy minimization in natural systems. Ideally, we will be able to map parameter regions in a $\left(\varepsilon_{0},\varepsilon_{1}\right)$ phase space for free energy-minimized systems with corresponding topographies. To illustrate this path, we briefly identify some salient work describing each of the three topographic types.

#### 3.3.1. Rich Club Topographies

One of the most well-known neural topographic structures has been described as being a *rich club*, characterized by clusters in which every node is connected with every other node in that cluster. Recently, Kim and Min (2020) take a graph theoretical approach to describing the rich-club structure in the brain [44].

Csigi et al., in their studies of rich clubs in complex networks, note that “the literature contains only a very limited set of models capable of generating networks with realistic rich club structure” [45]. They propose a means for generating rich club structures. One of the future directions that can ensue from this present study will be to create a means for generating certain specific topology types, such as rich clubs, within the context of 2-D CVM free energy minimization.

#### 3.3.2. Small-World Topographies

Watts and Strogatz (1998) were the first to introduce the notion of a “small-world” network, identifying that between completely regular and completely random topographies, there was a topographic type that was both highly clustered and at the same time, had a short path length between nodes [16]. (For our discussion, this would be a short path length between active nodes, or those in state **A**.) They noted that this topography allowed networks to have enhanced signal-propagation speed, computational power, and synchronizability. Muldoon et al. have developed a metric to assess what they term to be “small-world propensity,” or the degree to which a system is likely to exhibit small-world-ness [46].

Various researchers have found that certain structural networks in the brain exhibit small-world characteristics. For example, Achard et al. have concluded that “correlated, low-frequency oscillations in human fMRI data have a small-world architecture that probably reflects underlying anatomical connectivity of the cortex” [47]. Iturria-Medina et al. (2008), using diffusion-weighted MRI, found that anatomical brain regions exhibited small-world characteristics [48].

Liao et al. provide a concise summary of key measures useful in evaluating small-world-ness [49], and Zhu et al. take a graph theory-based to investigate the topological organization of the brain connectome, with particular attention to small-world properties (2020) [50].

Friston, describing how perturbations can influence a neural system, notes that “The emerging picture is that endogenous fluctuations are a consequence of dynamics on anatomical connectivity structures with particular scale-invariant and small-world characteristics,” and that these models “… also speak to larger questions about how the brain maintains itself near phase-transitions (i.e., self-organized criticality and gain control)” [51].

The notion of “small-world-ness” uses the notion of *geodesic distance*, where between any pair of nodes in an unweighted network, one can calculate the geodesic distance, which is given by the minimum number of edges that must be traversed to travel from the starting node to the destination node. In particular, a network is said to be a *small-world network* (or to satisfy the small-world property) if the mean geodesic distance between pairs of nodes is small relative to the total number of nodes in the network. Watts and Strogatz further introduced the notion of a global clustering coefficient, *C*, that is given as the ratio of three times the number of triangles over the number of connected triples. One potential direction will be to correlate the global clustering coefficient with measures that can be obtained from nodes within a 2-D CVM grid.

#### 3.3.3. Scale-Free Topographies

A final topography type to which we can investigate the application of the 2-D CVM are *scale-free topographies*.

The essential visual notion underlying a scale-free topography is that the distribution of patterns, in terms of the their relative scale, is independent of the particular scale with which we examine the topography. A fractal pattern is scale-free, and so is a snowflake.

Scale-free topographies can be found in nature, e.g., in natural landscapes. The notion of scale-free has also been used to characterize both functional and structural networks in the brain. (See, e.g., Yao et al. (2015) [52].) Studies by Eguíluz et al. (2005) identify an exponentially-truncated power law dynamic in macroscale functional MRI (fMRI) data [53].

The notion of scale-free behavior is generally linked to power-law dynamics, especially with regard to neural dynamics.

While initially, scale-free systems were thought to be ubiquitous, more recently, various researchers have challenged that assertion. As an example, Briodo and Clauset (2019) analyzed nearly 1000 networks, from across different domains, and found that relatively small proportions followed any real degree of power-law behavior [54].

Zhou et al. (2020), however, define a fundamental property-the degree-degree-distance-which is possessed by each link in a system. The distribution of this property can follow a power law, and can be used to characterize whether or not a system follows power law dynamics, and thus, whether it would be classed as being “scale-free” [55]. They find that, using this metric, the preponderance of scale-free systems (following a power law) is actually substantive. They find that their model “… predicts that the degree–degree distance distribution exhibits stronger power-law behavior than the degree distribution of a finite-size network, especially when the network is dense.” They further identify how distributions change as networks evolve into more dense stages.

One of the important aspects of describing scale-free, or more particularly *fractal* patterns, is selection of a measure for the fractal nature of a given system. Early work, e.g., by Mandelbrot [56], emphasized the notion of statistically self-similar constructions. As a second generation of fractal descriptions, Jiang and Yin (2014) introduced the *ht-index*, or *head-tail index*, to quantify the scaling or fractal structure of geographic features [19]. The *ht-index* essentially identifies the number of hierarchical levels in a fractal system. The *ht-index* is easy to interpret, however, it is not sensitive to the evolution of a fractal [57].

Recently, Gao et al. (2017, 2019) introduced the UM1 and and UM2 measures, where UM denotes “unified metrics” [57,58]. The UM1 metric is based on the Cumulative Growth Rate, or CGR, index (Gao et al., 2016) [59]. The UM2 metric is based on the Rato of Areas, or RA, index [57], which is an alternative to the CGR index in addressing the insensitivity issue of *ht-index*, and measures the heavy-tailed scaling of “far more small things than large ones.” In combination, the UM1 and UM2 metrics can offer a more sensitive means of modeling a topography’s fractal nature.

The value of using a combined UM1/UM2 approach would be that the *ht-index*, which is well-adopted in the community studying fractal properties, does not address the evoluton of fractals. In contrast, one of the primary investigation goals, after basic phase space mapping is complete, will be to map trajectory formulation of the configuration variables as there is a trajectory evolution within the $\left(\varepsilon_{0},\varepsilon_{1}\right)$ phase space. We would also want a means for describing topographic evolution, concommitant with the change in configuration variables.

Thus, the challenge, which is one of the next steps in creating a correlation between grid patterns resulting from 2-D CVM free energy minimization and actual topographies, will be to identify how the various fractal measures (e.g., UM1 and UM2) can be applied to 2-D CVM patterns. This will be addressed in later works.

The particular value of this next investigation will be that, to the extent that we can characterize how small change in the enthalpy parameters (ε0,ε1) influence the configuration values, and then identify the next correlation–between the changes in the configuration variables and the fractal measures–then there is an opportunity to not only track fractal evolution in a 2-D system, but also to potentially predict the next evolutionary stage, if we can identify a correlation betwen a trajectory in the enthalpy paramters and fractal evolution.

## 4. Materials and Methods

This section describes the methods for designing an initial grid pattern and for choosing an *h-value* that will guide free energy minimization.

### 4.1. Method Overview

The same basic codes, with minor variations for specific experiments, have been used throughout this work. The delicate and time-consuming aspect, requiring the greatest attention to verification and validation, has been computing the configuration variables. Once the configuration variables are obtained, it is straightforward to compute the free energy and other thermodynamic variables.

The overall method for achieving a free energy-minimized pattern, starting with an initial pattern, is in three steps. (This is the process for grid patterns where the activation enthalpy is zero ($x_{1}=x_{2}=0.5$), and the analytic solution therefore provides a starting point.)Devise an initial pattern, and compute the configuration variables associated with this initial pattern,Using this initial set of (not-at-equilibrium) configuration variables (specifically, the set of *interpretation variables*, $z_{1}$, $z_{3}$, and $y_{2}$), identify an appropriate *h-value* that reasonably corresponds to that set of interpretation variables, using both graphical and tabular depictions of the configuration variables as a function of the *h-value*, and thenUse that *h-value* as the key parameter in the code that modifies the pattern until a free energy minimum has been achieved. This new resulting pattern exemplifies how the initial pattern will appear after free energy minimization, and the corresponding set of configuration variable values identify a position in the phase space, associating the selected *h-value* with a set of configuration variable values.

This procedure of estimating a candidate *h-value* is needed, because we cannot modify the *h-value* itself during the course of the free energy minimization. The reason for this is that increasing the *h-value* always lowers the enthalpy (see Figure 7), and thus, if the code allowed the *h-value* to be adjusted, the system would keep increasing the *h-value* in order to achieve progressively lower free energy values through *h-value* lowering and not via topography modifications.

Once an *h-value* has been selected, free energy minimization is achieved through a simple strategy of looping through a control sequence where two nodes are randomly selected (one in state **A** and another in state **B**), and swapping the activations of the two nodes. If the resulting free energy is at a lower value than previously, the swap is kept. Otherwise, the prior activation states are returned to the nodes. This process is continued until changes in the free energy value are under a certain small threshold, or a limit on the total number of swaps is reached.

It should be noted that the code computing the configuration variable values is exact for the $x_{i}$, $y_{i}$, and $w_{i}$, and is approximate for the $z_{i}$, in that the $z_{i}$ values are computed for the $z_{i}$ triplets that can be counted horizontally (within a row above or below the node in question). The $z_{i}$ values that need to be counted vertically (that is, not extending to two rows above or below the node in question) were not included in this original code set. Detailed hand-calculations (on small-scale examples), as well as comparisons of theoretical vs. computational for larger samples, showed that the results obtained with this approximation yielded the same values as both the theoretical and the full (manually-counted) computational values.

Ongoing work, transitioning the code from a straightforward structural approach to object-oriented Python, will incorporate the full computation of the $z_{i}$.

Appendix A describes the codes, their documentation, and verification and validation (V&V) efforts.

### 4.2. Method Illustration

The three steps identified above are illustrated in the following example figures.

#### 4.2.1. Step 1: Compute Initial Configuration Variable Values

Figure 15 shows an initial “extreme rich club-like” pattern, in which the “on” (state **A**) nodes are massed together. (Recall that the grid shown depicts a wrap-around envelope in both directions, so the apparent set of two masses of **A** units are actually connected with each other “behind” the grid, and are also connected to themselves top and bottom.)

As shown in this figure, the first step is to compute the total numbers of each of the configuration variables. This was done both manually and computationally, in order to provide further code verification. The values for the total $Z_{i}$ are shown in the columns on the right-hand-side of Figure 15. (Recall that the $Z_{i}$ count is being done for the horizontally-oriented $Z_{i}$’s only, but that the values brought into the equations as $z_{i}$’s have been scaled appropriately.)

#### 4.2.2. Step 2: Identify Approximate *h-Value*

For step 2, the initial values for $z_{1}$, $z_{3}$, and $y_{2}$ obtained are compared with the analytically-derived values for these variables for certain *h-value*. This is shown in Figure 6, for two cases: (1) the “extreme rich club-like” grid (right-hand-side) that is used as a process example in this section, and also (2) a “scale-free-like” pattern (whose configuration variable values are depicted in the center of the figure). (Experiments on the “scale-free-like” pattern were described in the previous Section 2.5.1).

As is evident from Figure 6, the configuration variable values for a given initial grid pattern typically do not correspond with a single *h-value*, meaning that the initial (manually-designed) patterns are not at equilibrium.

The implication of this initial not-at-equilibrium state is that we can choose an *h-value* that seems reasonable for a given initial state, and perform free energy minimization on that pattern using the selected *h-value*.

At the beginning of this work, the working hypothesis was that the resulting configuration values, once a system had been brought to free energy equilibrium, would correspond to the analytically-predicted values. This turned out to not be the case. (See a more detailed discussion in Appendix A).

For example, if we take a nominal *h-value* of 1.65, the “extreme rich club-like” pattern’s initial value for $z_{1}$ is 0.418, which is slightly above the analytic prediction for $z_{1}$ at h=1.65. (See Table 2 for a summary of numeric values.) Overall, the initial configuration values for the “extreme rich club-like” pattern coalesce neatly around a single *h-value*.

For other initial patterns, the corresponding initial set of configuration variable values may be more dispersed, as was the case for the “scale-free-like” pattern. In that case, the course of action is to select a target *h-value* that is between two of the configuration variable values that are close, with a view to bringing other configuration variable value(s) closer into alignment with a topography indicated by the first two configuration variable values.

#### 4.2.3. Step 3: Perform Free Energy Minimization

Surprisingly, the actual configuration variable values after free energy minimization differed strongly from the analytically-predicted values.

The resulting pattern (shown in Figure 16) was achieved after minimizing the free energy for an initial “extreme rich club-like” grid, where h=1.65. (For details, see Maren (2019a) [4]).

As a contrast to the “scale-free-like” grid configuration used in Figure 11, this pattern had only one large compact region of nodes in state **A**, which was wrapped-around the grid envelope, as shown in Figure 15. This configuration was designed to maximize the number of pairwise and triplet configurations that put “like-near-like.”

Once the target *h-value* was selected, the computational process of swapping randomly-selected **A** and **B** nodes continued until a user-defined swap-limit was reached, or the free energy did not further decrease.

### 4.3. Method for the Case Where the Interaction Enthalpy Is Zero

When the interaction enthalpy is zero ($\varepsilon_{1}=0$), then we can find the configuration variable values associated with any reasonable value of $\varepsilon_{0}$. This is done very simply with an analytic solution. Details are presented in full in Maren (2019a) [4]. Relevant code is identified in Appendix A.

## 5. Conclusions

Free energy minimization of a 2-D cluster variation method (CVM) grid pattern induces a resulting pattern in which the characteristic configuration variable values (nearest-neighbor, next-nearest-neighbor, and triplets) are associated with specific values for activation and interaction enthalpy parameters. In this work, we have identified various configuration variable value sets at points along each of the two axes; where the activation enthalpy is zero (and an analytic solution can be achieved), and where the interaction enthalpy is zero (and a straightforward analytic solution is possible). We present a computational means for performing free energy minimization, and thus obtaining the actual configuration variable values at an enthalpy-parameter-specified state.

The resulting topographies show interesting properties, such as connectivity between regions of node clusters. Future work will advance the free energy minimization methods and further specify the phase space by associating sets of configuration variable values $\left(z_{1},z_{3},y_{2}\right)$ with enthalpy parameters $\left(\varepsilon_{0},\varepsilon_{1}\right)$, and will also characterize topographies associated with various phase-space regions.

## Figures and Tables

**Figure 1 entropy-23-00319-f001:**
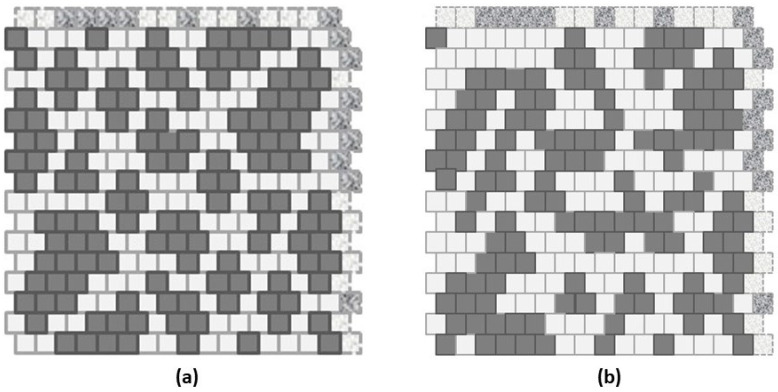
Illustration of (**a**) a manually-designed 2-D CVM grid that is (**b**) moved toward (but not yet reaching) a free energy equilibrium configuration for h=1.65, where $h=exp(2\varepsilon_{1})$.

**Figure 2 entropy-23-00319-f002:**
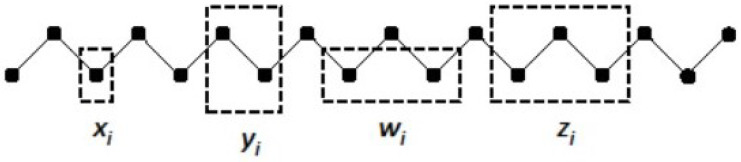
The 1-D single zigzag chain is created by arranging two staggered sets of *M* units each. The configuration variables shown are $x_{i}$ (single units), $y_{i}$ (nearest-neighbors), $w_{i}$ (next-nearest-neighbors), and $z_{i}$ (triplets).

**Figure 3 entropy-23-00319-f003:**
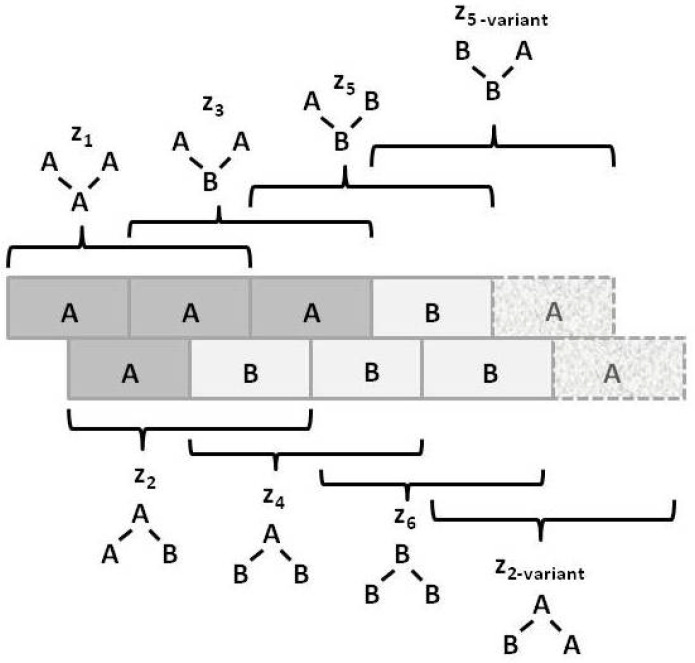
The six ways in which the configurations $z_{i}$ can be constructed.

**Figure 4 entropy-23-00319-f004:**
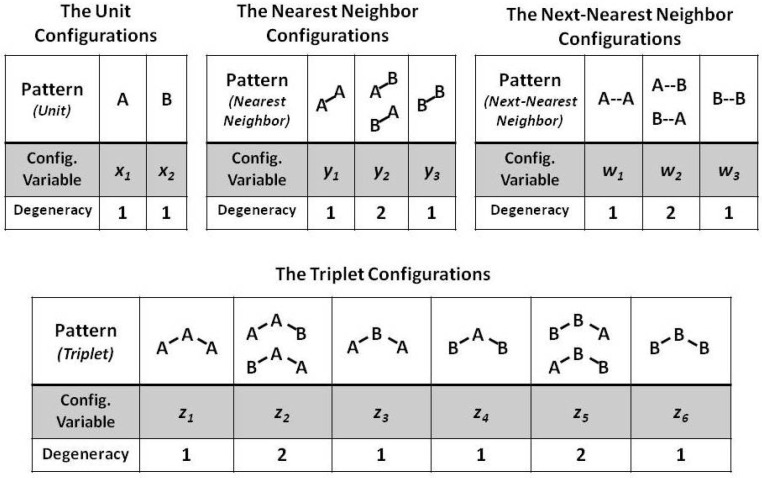
Illustration of the configuration variables for the cluster variation method, showing the ways in which the configuration variables $y_{i}$, $w_{i}$, and $z_{i}$ can be constructed, together with their degeneracy factors $\beta_{i}$ and $\gamma_{i}$.

**Figure 5 entropy-23-00319-f005:**
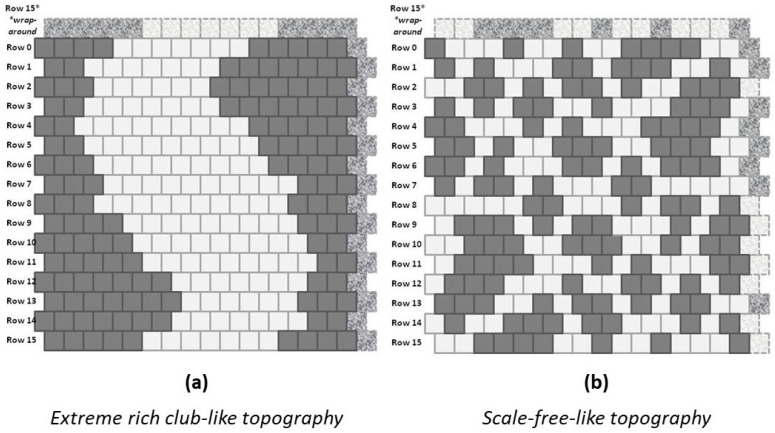
Illustration of the two different grids for experiments with the 2-D CVM system; (**a**) an “extreme rich club-like” topography is composed of a single “landmass” of **A** units, wrapping around the grid both vertically and horizontally, and (**b**) a “scale-free-like” topography was constructed from a series of small “landmasses” and “islands” of **A** units; see Section 2.5 and Section A.6 for methods. The mottled grey units (both light and dark) on the upper and far right of the grids represent the first edge of the vertical and horizontal wrap-arounds, respectively.

**Figure 6 entropy-23-00319-f006:**
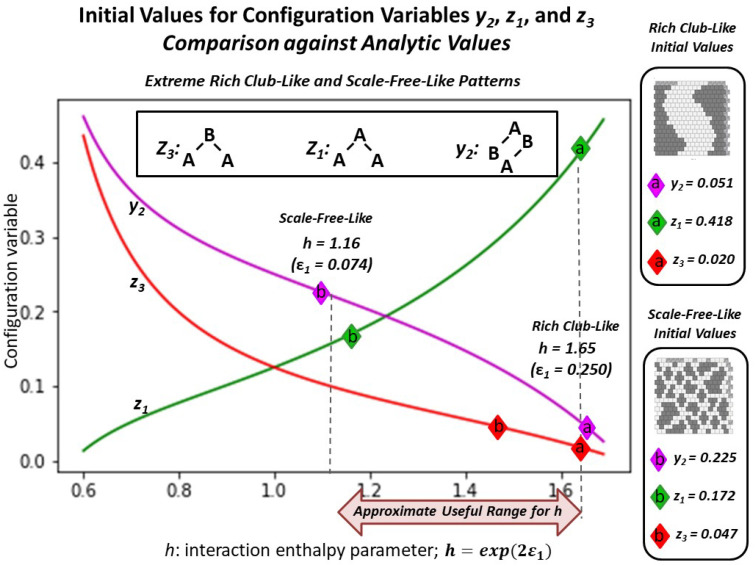
The initial configuration variable values for $z_{1}$, $z_{3}$, and $y_{2}$ are mapped against the analytically computed values for each of two cases, each containing an equal number state **A** and state **B** nodes (128 nodes each). Configuration variable values for an initial “extreme rich club-like” system (on the right-hand-side) are closely aligned with h=1.65, with diamonds indicating the initial configuration variable values denoted “**a**.” Configuration variable values for a “scale-free-like” system (in the center) has two of its configuration variable values ($y_{2}$ and $z_{1}$) closely surrounding h=1.16. The third configuration variable value for the “scale-free-like” system, $z_{3}$, lies far to the right. The diamonds denoting these configuration variable values are denoted “**b**.” See the text for further discussion.

**Figure 7 entropy-23-00319-f007:**
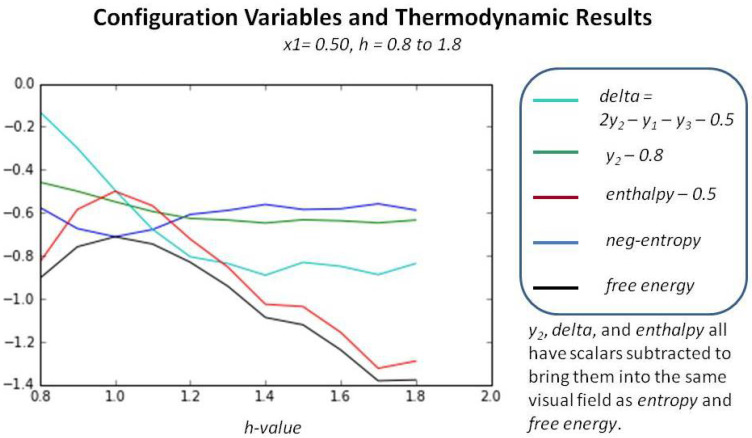
Configuration variable and thermodynamic values for the case where $x_{1}=x_{2}=0.5$, and where the interaction enthalpy parameter *h* ranges as h=0.8…1.8, with step size = 0.1. Results are illustrative, taken from one trial (of many) where the initial grid was randomly generated, and the final configuration variable and thermodynamic values were noted after the system had been either been brought to an equilibrium state or a predefined limit on swap attempts had been reached. Section A.4 contains detailed material relating to this figure.

**Figure 8 entropy-23-00319-f008:**
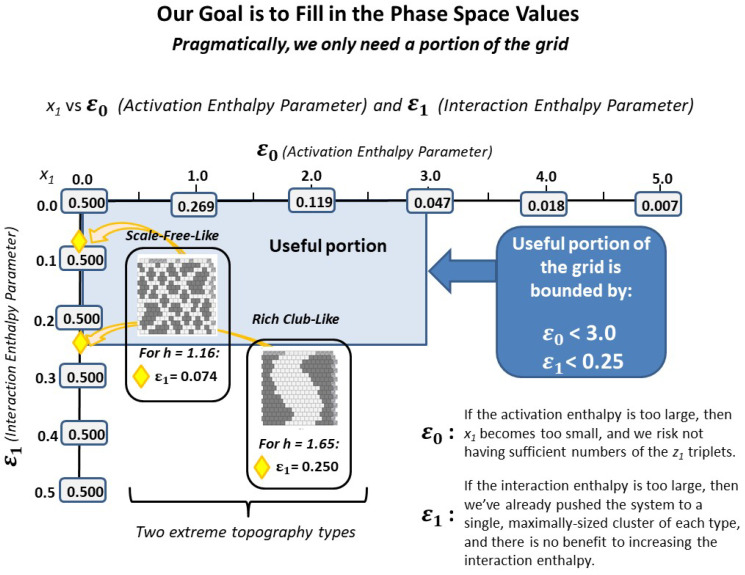
The useful phase space region for a 2-D CVM.

**Figure 9 entropy-23-00319-f009:**
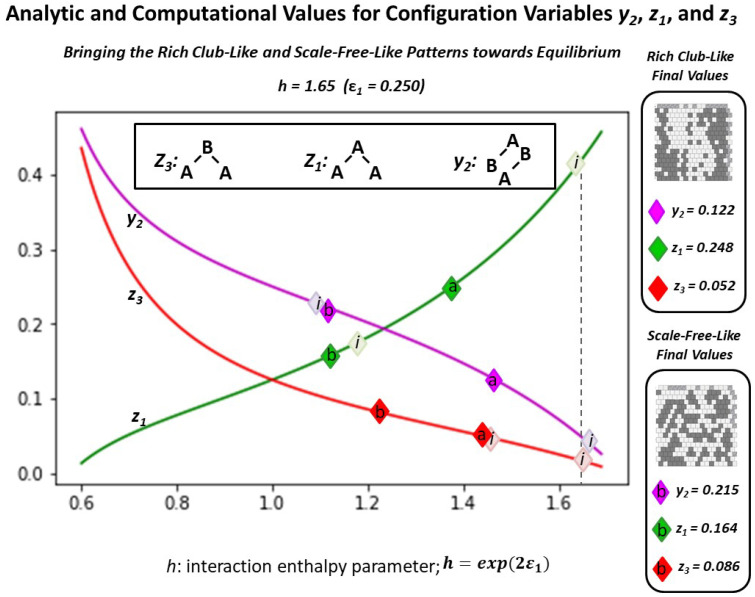
Configuration variable values after free energy minimization for two different initial patterns, with the *h-value* set to 1.65 for each case; **a**. Results for the “extreme rich club-like” pattern, and **b**. results for the “scale-free-like” pattern, where both initial patterns were presented in Figure 5.

**Figure 10 entropy-23-00319-f010:**
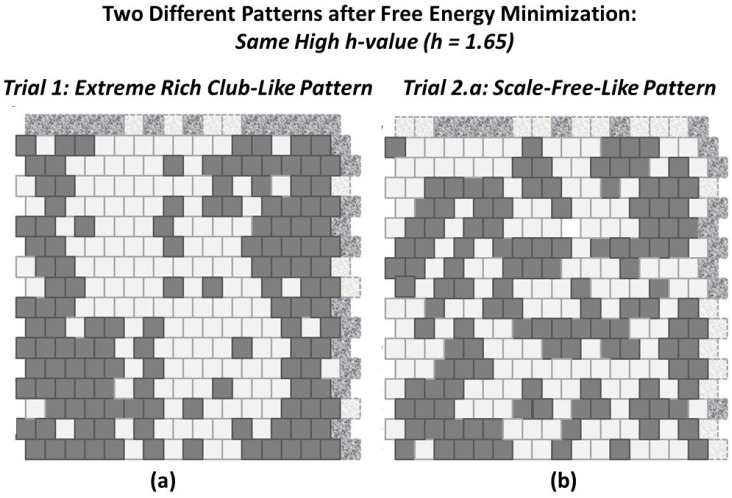
Results after free energy minimization for two different initial patterns, with the *h-value* set to 1.65 for each case; (**a**) Results for the “extreme rich club-like” pattern, and (**b**) results for the “scale-free-like” pattern, where both initial patterns were presented in Figure 5.

**Figure 11 entropy-23-00319-f011:**
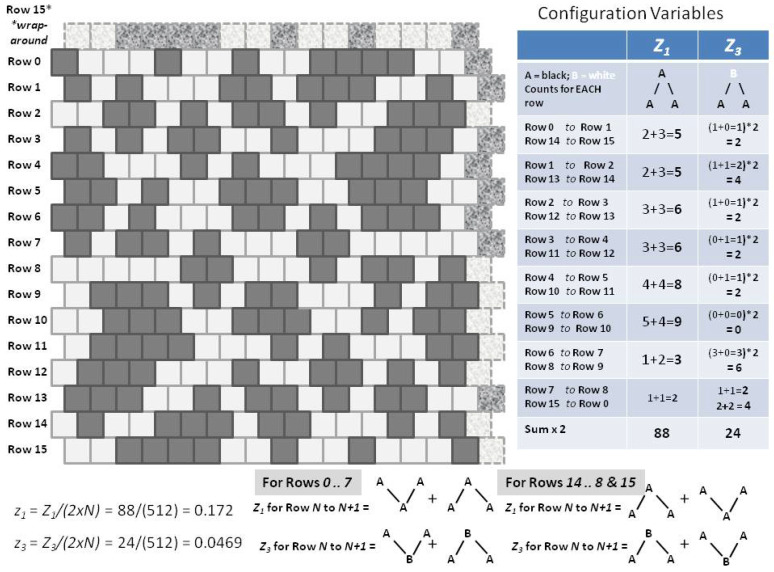
A 2-D CVM “scale-free-like” system with an equal number state **A** and state **B** nodes (128 nodes each). This grid pattern was initially presented in Figure 5b.

**Figure 12 entropy-23-00319-f012:**
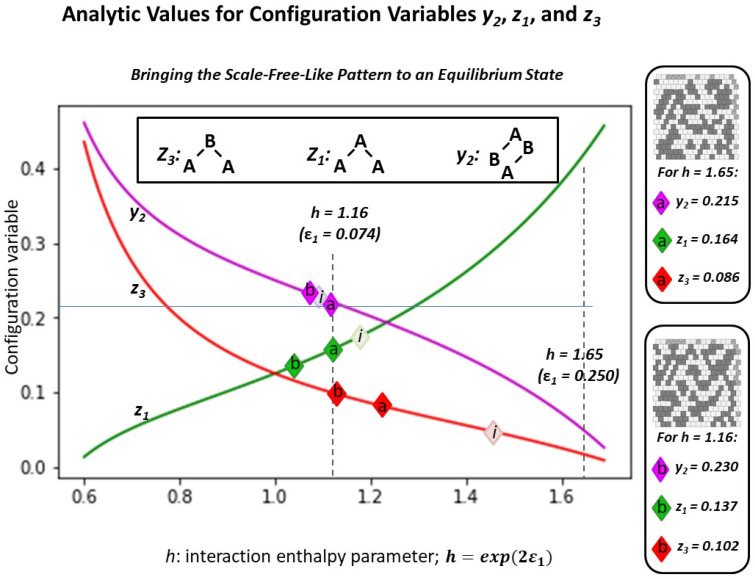
The 2-D CVM “scale-free-like” system is brought closer to an equlibrium state for each of two *h-values*: (a) *h* = 1.65, and (a) *h* = 1.16. The resulting configuration variable values ($z_{1}$, $z_{3}$, and $y_{2}$) are shown denoted “**a**” and “**b**,” respectively. The initial configuration values are shown in the lighter-shaded diamonds, marked with an *i*.

**Figure 13 entropy-23-00319-f013:**
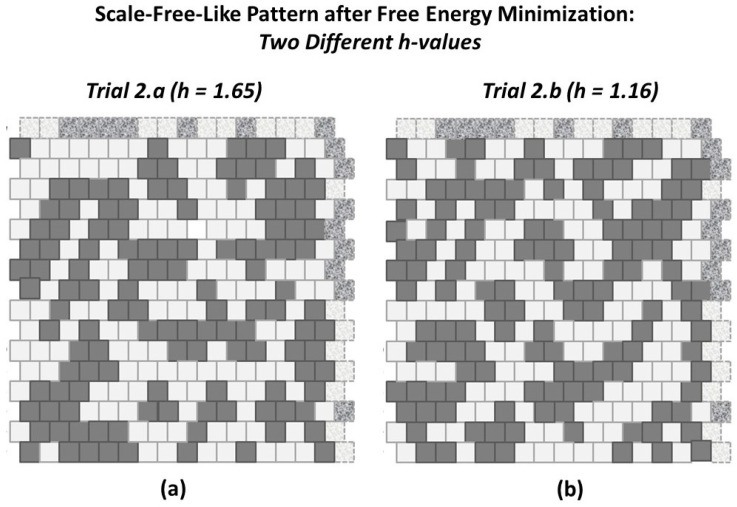
A 2-D CVM “scale-free-like” system after being brought to a free energy minimum, with two different *h-values*; (**a**) h=1.65, and (**b**) h=1.16. Note that there are still an equal number state **A** and state **B** nodes (128 nodes each).

**Figure 14 entropy-23-00319-f014:**
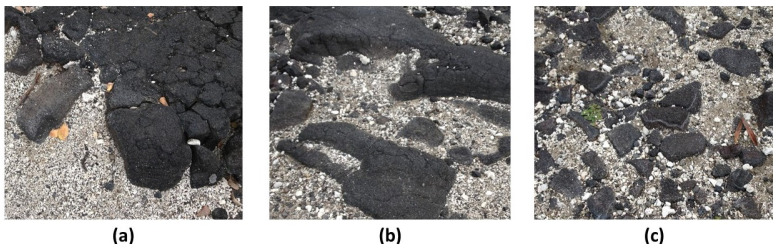
Illustration of three naturally-occuring topographies, observed within several yards of each other on the 1871 Trail in the Pu‘uhonua o Hōnaunau National Historical Park near Captain Cook, on the Big Island of Hawai‘i: (**a**) Larger lava rocks surrounded by white coral sand, where all lava rock elements are connected with each other. (**b**) A more fragmented version, in which various distinct sizes of lava rocks appear, yielding a range of lava rock sizes and varied degrees of connectiveness. (**c**) A yet more granular version of the same, with coral fragments approaching the size of some of the lava pebbles. Photos by A.J. Maren, 2020.

**Figure 15 entropy-23-00319-f015:**
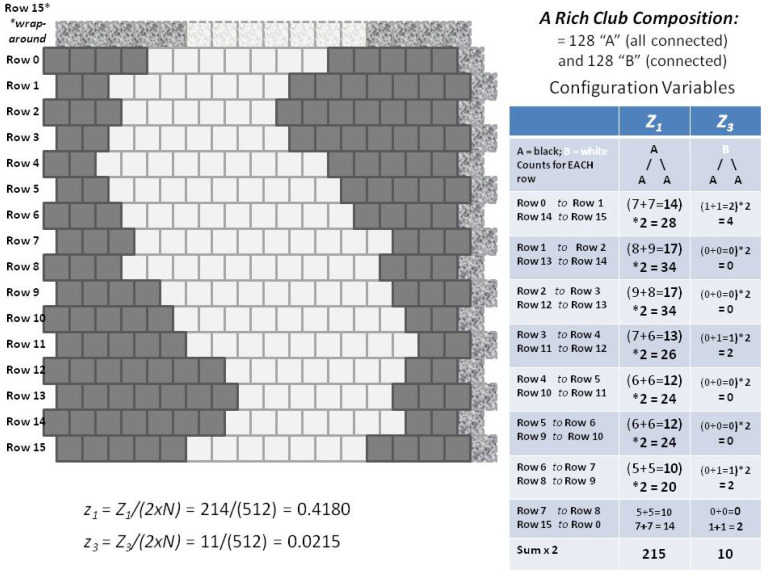
A 2-D CVM “extreme-rich-club-like” system with an equal number state **A** and state **B** nodes (128 nodes each). This system was originally introduced in Figure 5.

**Figure 16 entropy-23-00319-f016:**
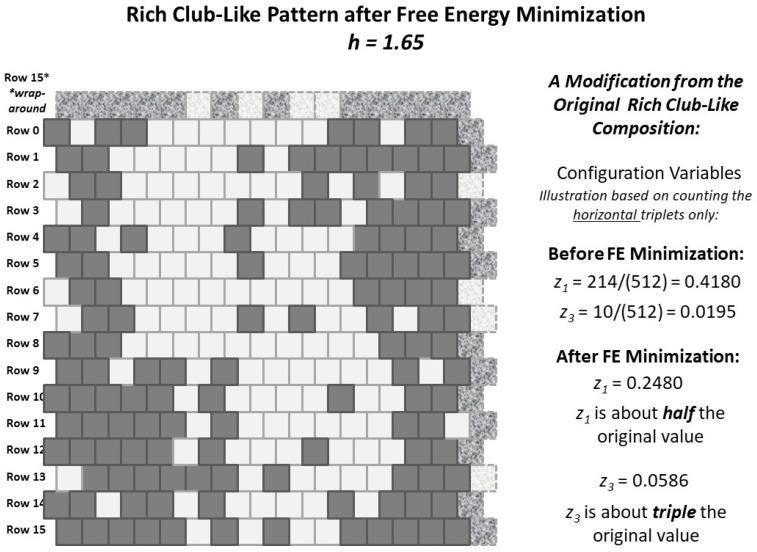
A 2-D CVM “extreme rich club-like” system after being brought to a free energy minimum. Note that there are still an equal number state **A** and state **B** nodes (128 nodes each). The original pattern for this grid was presented in Figure 5a.

**Table 1 entropy-23-00319-t001:** Glossary of thermodynamic terms.

Variable	Meaning
Activation enthalpy	Enthalpy $\varepsilon_{0}$ associated with a single unit (node)
β	$\beta =1/k_{\beta }T$; β can be set to 1 for our purposes
Configuration variable(s)	Nearest neighbor, next-nearest neighbor, and triplet patterns
Degeneracy	Number of ways in which a configuration variable can appear
Enthalpy	Internal energy *H* results from both per unit and pairwise interactions
Entropy	Denoted *S*; the distribution over all possible states
Equilibrium point	By definition, the free energy minimum for a closed system
Equilibrium distribution	Configuration variable values when free energy is minimized for given *h-value*
Ergodic distribution	Achieved when a system is allowed to evolve over a long period of time
Free energy	*F* = *H-TS*; sometimes *G* (Gibbs free energy) is used instead of *F*
*h-value*	$h=e^{2\beta \varepsilon_{1}}$, where $\beta =1/k_{\beta }T$
Interaction enthalpy	Between two unlike units, $\varepsilon_{1}$; influences configuration variables
Interaction enthalpy parameter	Another term for the *h-value* where $h=e^{2\varepsilon_{1}}$
$k_{\beta }$	Boltzmann’s constant
Temperature	Temperature *T* times Boltzmann’s constant $k_{\beta }$ is set equal to one

**Table 2 entropy-23-00319-t002:** Select configuration variable values and thermodynamic values when the *h-value* is set to 1.65 for each of two patterns brought towards free energy minimization; the “extreme rich club-like” pattern and the “scale-free-like” pattern. For comparison, the analytic solution variable values are also given.

Variable	Scale Free Initial	Scale Free Cmpt-Equil	Rich Club Initial	Rich Club Cmpt-Equil	Analytic Equil *
$z_{1}$	0.1719	0.1641	0.4180	0.2480	0.4209
$z_{3}$	0.0469	0.0859	0.0195	0.0566	0.0664
$y_{2}$ *	0.2246	0.2148	0.0508	0.1543	0.0477

* Values for *y*_2_ reported here have been divided by two from that given in the experimental results, to account for the degeneracy factor of two contributing to the total *y*_2_ counts. Values for the analytic solution are approximated between those for *h-values* of 1.6 and 1.7.

## Data Availability

The data used to define the two initial grid patterns used in this study are contained within the codes for performing free energy minimization on pre-defined patterns, identified in *Sample Availability*. The mechanism for generating randomly-defined grids, used for obtaining thermodynamic quantities described in Figure 7, is identified in codes available, see *Sample Availability*.

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
