# Peer review of "The 2-D Cluster Variation Method: Topography Illustrations and Their Enthalpy Parameter Correlations"

_entropy, 2021, doi:10.3390/e23030319_

Round 1

Reviewer 1 Report

I enjoyed reading this treatment of the 2D cluster variation method. I thought you took the reader through the numerical analyses carefully and clearly. I think this contribution will have more impact if you included a couple of orientating paragraphs in the introduction – and attended to a couple of minor details. Perhaps you could consider the following:

Major points

I think you need to overview the numerical and analytic procedures described in the paper so that the reader knows what to expect. I would recommend something like the following in the introduction

“In what follows, we will use numerical analyses of synthetic activation patterns on a two-dimensional grid as they evolve to – and attain – an equilibrium steady-state distribution. The focus of these analyses is on the characterisation of various patterns in terms of configuration variables; namely, the statistics of local patterns of co-activations or dynamical motifs. By equipping configuration variables with an activation and interaction enthalpy, one can cast evolution to equilibrium as a free energy minimising process. To simulate this process, one can randomly exchange activation values between grid points and retain the switch if free energy falls. The focus of this paper is on the different kinds of patterns (i.e., rich club, scale free, et cetera) that emerge under different contributions of activation and interaction enthalpy to free energy. In summary, will use worked (numerical) examples to illustrate how different kinds of patterns emerge under the same free energy minimising formalism – and discuss how this may have implications for self-organisation and message passing on graphs, such as the brain."

In addition, I think you need to put your sentence about "”remaining sections will investigate this figure in more depth" before the figure. For example:

“The following figure illustrates the sort of characterisation afforded by the 2D CVM approach. The remainder of this paper will unpack the construction and interpretation of this figure – to provide a worked example of how the 2D CVM approach is applied."

In the closing discussion, it might be an idea to explain to the reader what steps would be necessary to apply the 2D CVM in practice. For example, could one apply this formalism to empirical timeseries and reverse engineer the activation and interaction enthalpies? Could the numerical analyses be used to generate dynamic graphs for Bayesian model selection or message passing schemes? And so on. In short, the numerical analyses presented in this paper give people a nice intuition as to the utility of the cluster variation method. It would be good to suggest practical applications – and what would need to be done in order to realise these applications.

Minor points

On line 124, you talk about active inference and emotion recognition. A more relevant reference might be the link between free energy minimisation and critical slowing – that has links to your discussion of scale free dynamics in the previous paragraphs (Friston, Kahan et al. 2014). In addition, you might be interested in related work on Markov blankets within the brain and the emergent scaling (power law) behaviour (Friston, Fagerholm et al. 2020).

On line 875, I would spell out "and V&V" (verification and validation)

In the materials and methods section, I would remove the first two paragraphs and put them in a software note at the end of the paper. You can then start with the Method Overview.

On line 877, what is “textith value”?

On lines 899 and 900, there appear to be some missing references ([?]).

I hope that these comments help should any revision be required.

Author Response

Thank you for your kind and thoughtful review, which has greatly increased the clarity of this work. I appreciate your suggestions that have opened up lines for future endeavors. Please see the attachment. 

Kind regards - AJM

Reviewer 2 Report

Title needs to be changed to reveal insight of the paper. The 2-D cluster variation method, by itself, has been published therefore it's misleading to refer to 'initial findings' particularly as there is no indication as to what the findings refer to.

Abstract: The abstract contains a number of statements that are not actually informative. For example, rather than state 'produce interesting results', indicate what the results are! The abstract fails to provide an understanding of the aim of the paper. It starts with a statement regarding 2-D image topographies and ends with a speculation regarding neural networks and their extension in the time domain.

The motivation for the paper starts with gratuitous statements that are not particularly well motivated. For example: Statement line 29-30 is gratuitous given that how 'overarching' any framework is is in the eye of the beholder. Without knowing what would qualify as 'overarching' it is difficult for me to assess whether this statement is correct or not. Statement line 32 that 'we have not had an appropriate free energy function' is once again gratuitous. By what definition are current suggestions (which are not actually listed) not appropriate? What would make them appropriate?

More generally, there is a total lack of a precise, operational definition of what the problem is that the author seeks to address. The author implicitly provides a sense of what they are trying to achieve by listing the properties of their proposed method but that, in no way, does specify the problem at hand, and neither that it make it a recognisable characterisation of what the problem(s) is(are) in neuroscience. It is clear that the author is trying to advance uptake of their work and there may well be value in it but it has to start by being explicit about what is the research question being addressed. In other words, start with the 'what', not the 'how'.

The connection between the concept of free energy minimisation and critical dynamics is not particularly clear. The authors appear to suggest (line 96) that it might be an example of how free energy minimization plays its role. This suggests the authors believe one implies the other but this is not explained and instead the relationship seems to hinger on loose behavioural similarities (the existence of non-equilibrium steady states).

On line 131, the authors articulate a question "how does this process [i.e., free energy minimization] evidence in terms of activation patterns within the brain?" This question is fundamentally problematic because free energy minimization is a principle, not an implementation. Now, the authors appear to be systematically conflating structural-related concepts (e.g., 'rich club', 'small world' and 'scale free') with functional concepts (neural activation patterns). Even when the above structural features are applied to functional connectivity, they are still applied to a graph-representation of correlations between neural activity. There is total lack of clarity on this point which makes the interpretation of the methods/results near impossible. In fact, it would be essential for the authors to actually operationally describe what 'neural activation patterns' actually represent. Not once are they defined (the words only appear in abstract, introduction then up to section 1.3.3) but more importantly, in sections 1.3.1, 1.3.2 and 1.3.3, the reference to neural activation patterns is inappropriate because the papers the authors reference clearly deal with structural networks. 

I am sorry that on the basis of this total lack of clarity regarding the question and the confusion on the concepts that are being used, I am unable to proceed with reviewing the technical content of this paper as I simply do not understand what this paper is trying to achieve.

Author Response

Thank you for your thoughtful and insightful comments. You have been very helpful in guiding greater clarity. Please see the attachment. 

Kind regards - AJM

Reviewer 3 Report

It is difficult to feel the contribution of this manuscript after reading it title. Thus, I would suggest revising the title.  In addition, the structure of the manuscript is also strange; the introduction section is too long to have 12 pages!  This section should be shortened substantially.  The results section should not appear earlier than the methods section, unless this manuscript is prepared for other journals.

As stated in the abstract, the 2-D custer variation method (CVM) was proposed in 1951, but this method was investigated today with this manuscript. This fact makes me wonder why there is no similar works during the past 70 year? Also, Line 49 reads as follows: “the CVM approach has not received a great deal of attention”. However, no reasons have been provided in the manuscript for such a strange fact given that CVM has several advantages (Lines 36 – 39).

The authors stated that “one reason that this might be the case is that - up until now - we have not had an appropriate free energy function.” This statement should be made with citations.

Section 1.3.3 discusses scale-free topographies. The notion of “scale-free” is highly related to fractals.  Indeed, as noted in Line 193, a fractal pattern is scale-free. There are three definitions (or generations) of fractals. Which definition was adopted in this study? I guess the second definition was adopted. If this is the case, please discuss the possibility of adopting the third definition of fractals (characterized using ht-index or improved metrics such as UM1 and UM2).

Author Response

Thank you for your kind review. Please see the attachment for a detailed response. 

I particularly found your suggestions of incorporating more advanced measures for fractal systems helpful and interesting; they provide a useful direction for future work. 

Kind regards - AJM

Round 2

Reviewer 3 Report

All my concerns have been well addressed. Thanks!